# A neural network for modeling human concept formation, understanding and communication

Liangxuan Guo [1,2,3,8], Haoyang Chen [4,8], Yang Chen [1,3,8,9] ✉,
Yanchao Bi [4,5,6,7,9] ✉ & Shan Yu [1,2,3,9] ✉

A remarkable capability of the human brain is to form more abstract conceptual representations from sensorimotor experiences and flexibly apply them independent of direct sensory inputs. However, the computational mechanism underlying this ability remains poorly understood. Here we present a dual-module neural network framework, CATS Net, to bridge this gap. Our model consists of a concept-abstraction module that extracts low-dimensional conceptual representations, and a task-solving module that performs visual judgment tasks under the hierarchical gating control of the formed concepts. The system develops transferable semantic structure based on concept representations that enable cross-network knowledge transfer through conceptual communication. Model–brain fitting analyses reveal that these emergent concept spaces align with both neurocognitive semantic model and brain response structures in the human ventral occipitotemporal cortex, while the gating mechanisms mirror that in the semantic-control brain network. This work establishes a unified computational framework that can offer mechanistic insights for understanding human conceptual cognition and engineering artificial systems with human-like conceptual intelligence.

A unique feature of human language and thought, as pointed out by Ferdinand de Saussure in 1916[1], is the ability to use a 'signifier' (for instance, symbolic reference) to communicate about 'signified' referents that are physically absent. This capacity to decouple mental concepts from immediate sensory content allows humans to plan, simulate and represent information beyond the 'here and now'. However, the computational framework that enables neural networks to form such concepts—initially dependent on sensorimotor stimuli but later independent of them—remains elusive.

In humans, concept processing comprises two coupled capacities: concept formation, where high-dimensional sensorimotor experiences are compressed into lower-dimensional representational spaces[2–5], whose dimensionality typically ranges from twenty to several hundred[6–10]; and concept understanding, where these concepts are reactivated to reinstate sensorimotor states and flexibly combined[5,11–15]. For example, hearing 'last night's dinner' would elicit rich event-related imagery (Fig. 1a), enabling communication of meanings through symbols. This bidirectional process is essential for concept processing in

[1]Laboratory of Brain Atlas and Brain-inspired Intelligence, Institute of Automation, Chinese Academy of Sciences, Beijing, China. [2]School of Future Technology, University of Chinese Academy of Sciences, Beijing, China. [3]State Key Laboratory of Brain Cognition and Brain-inspired Intelligence Technology, Institute of Automation, Chinese Academy of Sciences, Beijing, China. [4]School of Psychological and Cognitive Sciences & Beijing Key Laboratory of Behavior and Mental Health, Peking University, Beijing, China. [5]IDG/McGovern Institute for Brain Research, Peking University, Beijing, China. [6]Institute for Artificial Intelligence, Peking University, Beijing, China. [7]Key Laboratory of Machine Perception (Ministry of Education), Peking University, Beijing, China. [8]These authors contributed equally: Liangxuan Guo, Haoyang Chen, Yang Chen. [9]These authors jointly supervised this work: Yang Chen, Yanchao Bi, Shan Yu. ✉e-mail: yang.chen@ia.ac.cn; ybi@pku.edu.cn; shan.yu@nlpr.ia.ac.cn

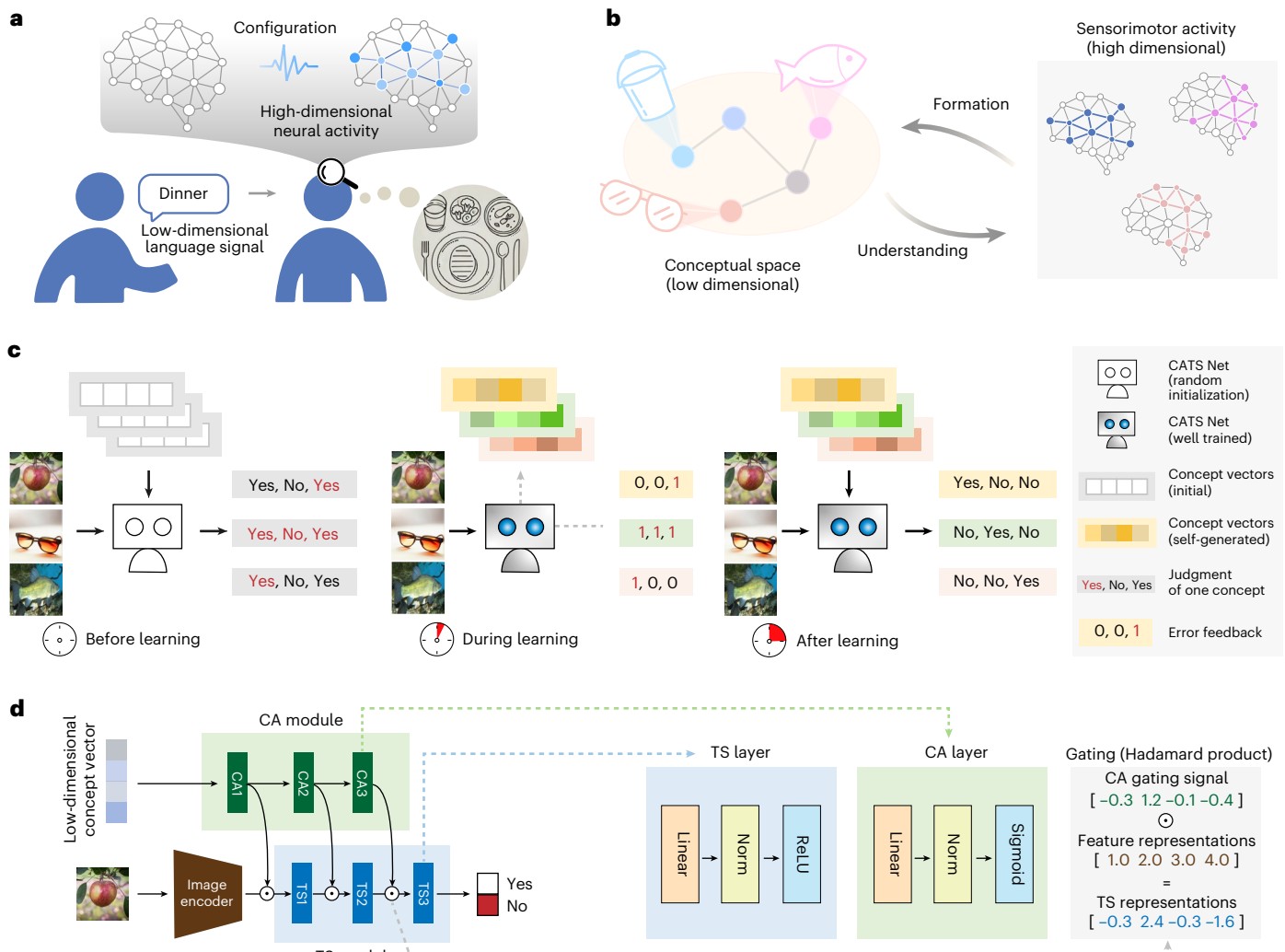

**Fig. 1 | Motivation, experimental approach and architecture of CATS Net for concept decoupling and formation. a**, The key characteristic of concepts is their decoupling from complex, high-dimensional sensorimotor information into lower-dimensional representations. For instance, the concept conveyed by a simple word like 'dinner' can evoke neural population activity patterns associated with dining scenes, even without direct sensory stimulation. **b**, A possible solution for concept formation is to compress sensorimotor neural circuits, independent of direct inputs, into low-dimensional representations. If these concepts can subsequently activate proper circuits to effectively accomplish the desired functions, it can be regarded as concept understanding.

**c**, Illustration of our CA task approach. After training from random CATS Net parameter weights and initial concept vectors (all same or totally random), the system gets a set of well-trained parameter weights and well-trained concept vectors, which further support successfully making binary judgment for a given image under a given concept. **d**, Schematic illustration of the dual-module architecture in CATS Net: the CA module receives low-dimensional conceptual inputs to generate a controlling signal for TS module; the TS module performs 'Yes/No' judgment according to sensory inputs and gating operation by the CA module. All images from PublicDomainPictures and Free-Images under a Creative Commons license CC0.

humans. A computational framework that simultaneously models both concept formation and concept understanding remains a key challenge in artificial intelligence and neuroscience.

Current approaches fall short of integrating these two functions. On the one hand, deep neural networks such as ResNet and vision transformers[16,17], or classic convolutional neural networks with attention modules[18,19], excel at learning representations, but entangle knowledge within millions of parameters, making it hard to decouple from the network or directly transfer to another agent. On the other hand, multimodal large language models[20–22] rely on pre-existing language symbols rather than modeling de novo concept formation from sensorimotor experience.

Here, inspired by a previously proposed network capable of flexible context-dependent processing[23], we propose CATS Net, a dual-module framework comprising concept-abstraction (CA) and sensorimotor task-solving (TS) modules (Fig. 1b). In this framework,

concept formation is modeled as CA forming a low-dimensional input space of concept vector, while understanding is modeled via a gating mechanism where concept vectors dynamically reconfigure the TS module.

We demonstrate that CATS Net can derive concepts from a visual binary classification task, and transfer concept knowledge between CATS Nets. Importantly, concept representations formed in CATS Net significantly correlate with human higher-order visual cortices, while the CA module aligns with the brain's semantic-control network[24], offering insights into the computational underpinnings of human conceptual processing.

## Results

### Unified modeling of concept formation and understanding
We introduced a concept-abstraction task, where CATS Net generates a series of highly compressed concept vectors corresponding to

particular visual category (Fig. 1c). Each learned vector functions as a functional classifier; for instance, an 'apple' vector configures the network to judge whether an image input to the TS module belongs to apple category.

This is achieved via a hierarchical gating mechanism where the CA module transforms low-dimensional concept vectors into layer-wise control signals that modulate the TS module's activity (Fig. 1d). The TS module is a multi-layer perception with a two-head classifier for binary (Yes/No) image judgment, to process features extracted by a pre-trained backbone. The framework's design is agnostic to the backbone architecture, demonstrating robust generalization across different structures such as ResNet50[16] and ViT-B/16[17] (Supplementary Fig. 1a).

The training process involves two phases: the network parameters learning phase, where the weights of CA module and TS module are trained together; and the concept-abstraction phase, where the concept vectors are updated. These two phases were executed in a round-robin fashion until identification accuracy plateaued. This two-phase training strategy, along with the selection of a pretrained ResNet50 as the feature extractor, a concept size of 20, and 3-layer CA and TS modules, was validated as optimal through ablation studies (Supplementary Fig. 1a). Using this established configuration here and for the rest of the current study, we trained 30 independently initialized models on the ImageNet-1k dataset[25], which successfully generated a set of visual concept vectors for task solving. For all unseen images from 1,000 categories tested on the ImageNet-1k dataset, the learned concept vectors for each category achieved a judgment accuracy ranging from 0.86 to 1.00, well above the chance level of 0.5 (Fig. 2a).

Furthermore, through visualization, we observed that the models indeed focus on the part of the input image that corresponds to the concept. Using class activation mapping (CAM)[26], with the same image input under different concept configurations, the network attends to different parts of the image (Fig. 2b). This shows that the network can adapt to different functions based on the conceptual input.

Importantly, our empirical comparisons indicate that the learnability of both the concept vectors and the CA and TS modules is equally critical. We compared our approach with 3 alternative methods for constructing a fixed concept space (Supplementary Fig. 1b): using (1) frozen 20-dimensional random vectors, (2) frozen Word2Vec vectors projected to 20 dimensions, or (3) 1,000-dimensional one-hot vectors.

The results revealed a crucial interplay between concept space learnability and network capacity. First, the trainable 20-dimension space significantly outperforms frozen and Word2Vec counterparts (random: mean difference 0.0192, 95% bootstrap confidence interval (CI) [0.0185, 0.0200] with 5,000 resamples, two-sided permutation test with 10,000 permutations, $P < 0.001$; Word2Vec: mean difference 0.0279, 95% bootstrap CI [0.0256, 0.0313] with 5,000 resamples, two-sided permutation test with 10,000 permutations, $P < 0.001$). These results suggest that imposing a fixed concept space would force the network to compensate for arbitrary mappings, a constraint that becomes severe under limited capacity. For instance, reducing the CA and TS modules from 3 layers to 1 layer dropped accuracy with a frozen random space from 0.944 to 0.793, whereas enabling concept learnability restored accuracy to 0.954.

Second, compared with the one-hot baseline, our approach was both more accurate and more scalable: a learnable 100-dimension concept space performed better (mean difference 0.0043, 95% bootstrap CI [0.0021, 0.0056], 5,000 resamples, two-sided permutation test with 10,000 permutations, $P = 0.0079$) Furthermore, one-hot codes preclude any semantic structure, and scale linearly with the number of classes, requiring redefinition of the space when new concepts are added.

### Semantic organization and human alignment of CATS Net
**Functional specificity of the concept space.** To investigate the properties of the emergent 20-dimensional concept space, we probed its

structure using the standard basis. Specifically, we used the set of 20 canonical one-hot vectors, which are referred to as the basis vectors throughout the rest of this paper. Then we tested the functional specificity of the basis vectors. We categorized the 1,000 classes of ImageNet validation set into 5 hyper-categories based on WordNet, and then counted the 'Yes' response of the input basis vectors to these hyper-categories. Unlike the uniform response observed before training (Fig. 2c), trained basis vectors showed higher selectivity to specific hyper-categories (Fig. 2d).

Taking a step further from the microscopic unit basis analysis to the macroscopic aspect, we introduce the 'functional entropy' to examine the overall functional specificity of the low-dimensional concept space. For a given concept vector, the functional entropy is computed over the number of 'Yes' response counts of all 1,000 classes (Fig. 2e and Methods). Low entropy corresponds to high functional specificity. We randomly sampled 1,000 points from the trained concept space; their entropy distribution was markedly lower than a random baseline (Fig. 2f), indicating a structured space with category-specific organization shaped by training.

**Representational similarity to human semantic models.** Next, we compared the CATS concept space with two complementary human semantic models: Binder65 (65 neurobiologically grounded dimensions[8]) and SPOSE49 (49 behavior-derived dimensions from THINGS similarity judgment[9]). Using a representational similarity analysis (RSA)[27] approach, we constructed representational dissimilarity matrices (RDMs) over 332 shared concepts and correlated CATS RDMs with the semantic model RDMs at both the average-RDM level and across 30 independently trained CATS instances.

As shown in Fig. 3a, the average CATS concept RDM showed significant correlations with Binder65 RDM (Spearman's $\rho = 0.14$, Mantel $P < 0.001$) and SPOSE49 RDM (Spearman's $\rho = 0.29$, Mantel $P < 0.001$). This correspondence was consistent across instances (Binder65, $t(29) = 18.28$, $P < 0.001$, Cohen's $d = 3.39$, 95% CI [0.03, 0.04]; SPOSE49, $t(29) = 25.43$, $P < 0.001$, Cohen's $d = 4.72$, 95% CI [0.06, 0.07]), indicating that CATS, despite being trained solely on visual categorization, generated a concept space similar to human conceptual organization. Notably, this similarity seems to be able to reflect the ability of CATS Net to capture abstract dimensions, as further evidenced by the significant correspondence with non-visual dimensions of Binder65 (for instance, spatial, temporal, emotional; Extended Data Fig. 1).

**Semantic interpretability.** To further explore the interpretability of our CATS Net concept space, we tried to provide semantic labels for our concept space dimensions. We used the SPOSE49 model as a reference framework, because it captures finer-grained, visually relevant features that have been validated to effectively explain human similarity judgment behaviors[9] and neural response patterns[28].

Specifically, we adopted a best-match procedure for each SPOSE49 dimension. Within each CATS instance, we computed Pearson correlations between a given SPOSE49 dimension and all concept dimensions of CATS Net, retaining the maximum correlation as the 'best match' score for that instance. This yielded, for each SPOSE49 dimension, a distribution of 30 best-match correlations across CATS instances. Figure 3b highlights four SPOSE49 dimensions (metal/tool, food, furniture and long/thin) for which almost all 30 CATS instances exceeded the nominal significance threshold ($P < 0.05$; $n = 334$ concepts), indicating robust convergence on similar semantic structure despite different random initialization. The complete results across all 49 dimensions are presented in Supplementary Fig. 2.

To directly visualize the structure of the formed concept space by CATS Net, we conducted identical task configuration training on a smaller-scale CIFAR-100 dataset[29]. Specifically, we got a set of 100 concept vectors by performing concept-abstraction task on CIFAR-100. Then, we performed hierarchical clustering to these vectors

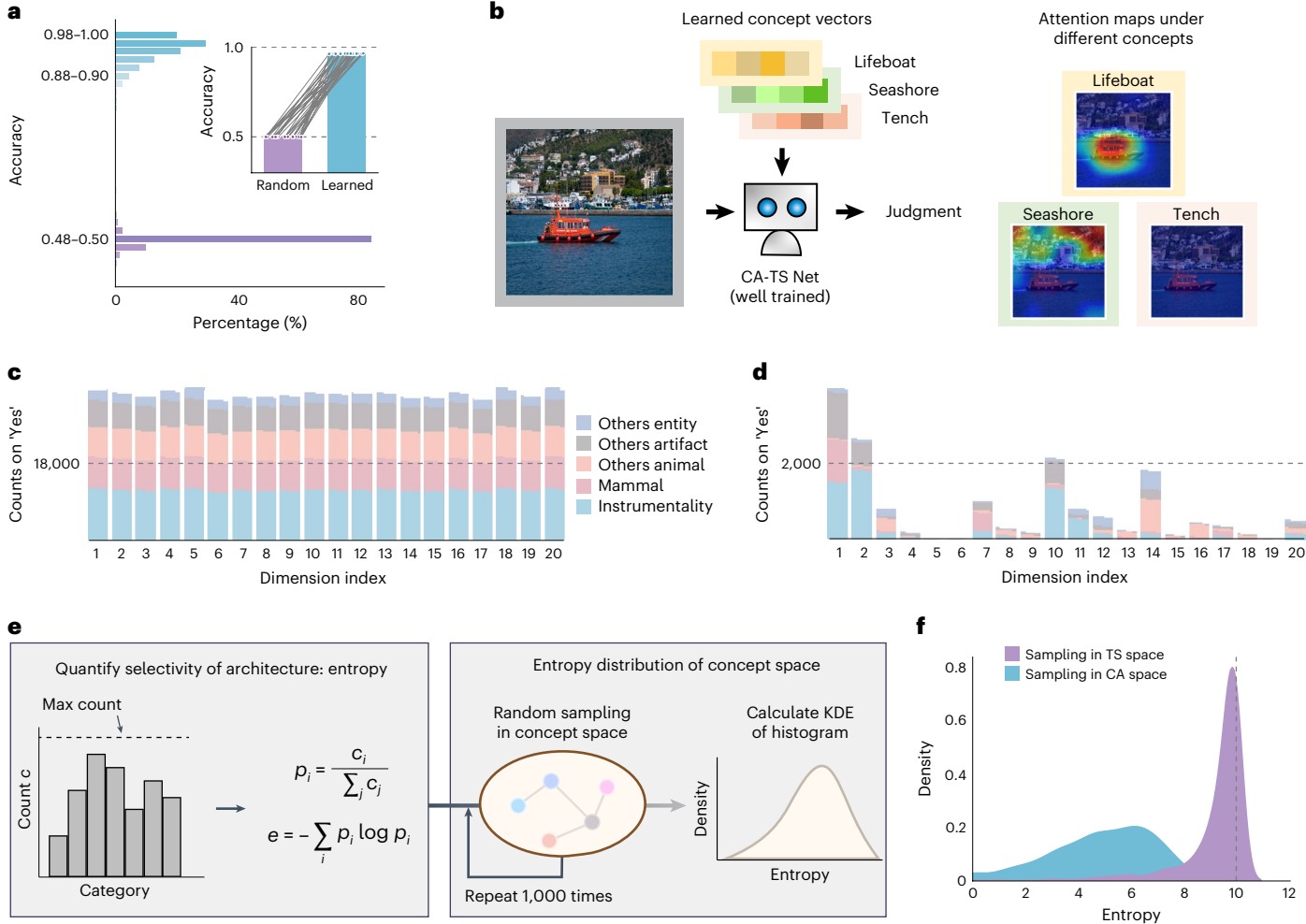

**Fig. 2 | Model performance and conceptual space semantic structure analyses. a**, The performance of concept abstraction by CATS Net on the ImageNet-1k dataset. The purple histograms depict the accuracy distribution of CATS Net for 1,000 initial concept vectors before learning and the blue ones are after learning. In the inset, the purple and blue bars represent the average of mean accuracy across 30 models before and after training, and each pair point represents the corresponding mean accuracy of each category. **b**, Visualization of selective attentions on the same input modulated by different concepts.

**c**,**d**, The functional specificity of the unit basis vectors on hyper-categories before (**c**) and after (**d**) learning. The height of bar indicates the number of 'Yes' response of the input basis vectors to these five hyper-categories. This result is a single case randomly chosen from 30 instances. **e**, Calculation pipeline of 'functional entropy', which quantitatively measures the functional specificity on the task. **f**, Probability density distribution of functional entropy in the trained concept space (blue) and task-solving parameter space (purple). All images from PublicDomainPictures under a Creative Commons license CC0.

based on cosine distance to visualize the internal structure of the low-dimensional concept space (Fig. 4a). This analysis reveals a modular structure characterized by distinct semantic clusters. Notably, semantically close categories formed clusters, for instance, clusters of people, animals, trees, fruit, furniture and automobiles. These concept vectors enabled the establishment of connections among concepts through diverse multidimensional relationships, including similarities in foreground shape (such as snakes and worms), foreground color (such as sweet pepper and sunflower), background (such as mushrooms and snail) and co-occurrence (such as palm tree, cloud and sea; tulip and butterfly).

### Communication by aligning the concept spaces of different CATS Nets

Next, we define a 'learning by communication' experiment to test whether capabilities could be transferred solely via low-dimensional concepts. Using the CIFAR-100 dataset, we ran 100 unique teacher–student pairs, each training the student network with one category held out. Each run comprised three phases: independent concept abstraction, concept alignment and concept transmission (Fig. 4c).

**Independent concept abstraction (phase 1).** In the first phase, an asymmetric training strategy was used to create a knowledge gap: the teacher network learned all 100 categories, while the student network learned 99, withholding one distinct category (for example, 'apple') per pair for transfer testing. Despite independent initialization, the emerging concept spaces showed significant structural similarity. For instance, the modular organization of the teacher's space (Fig. 4a) was mirrored in the student's space (Fig. 4b). Quantitative analysis using cosine-distance RDMs confirmed this alignment across all teacher–student pairs (Spearman's $\rho = 0.35$, $t(99) = 3.83$, one-tailed $P < 0.001$, Cohen's $d = 0.38$), indicating a shared internal structure for communication.

**Concept space alignment via a translation module (phase 2).** In the second phase, the two concept spaces were aligned with a translation module, that is, a neural network establishing a map from teacher concept space to student concept space, which was trained with expanded concept vectors (Methods). To verify whether semantic details were preserved during translation, we analyzed the layer-wise internal representations of the translation module. Visualization of RDMs revealed

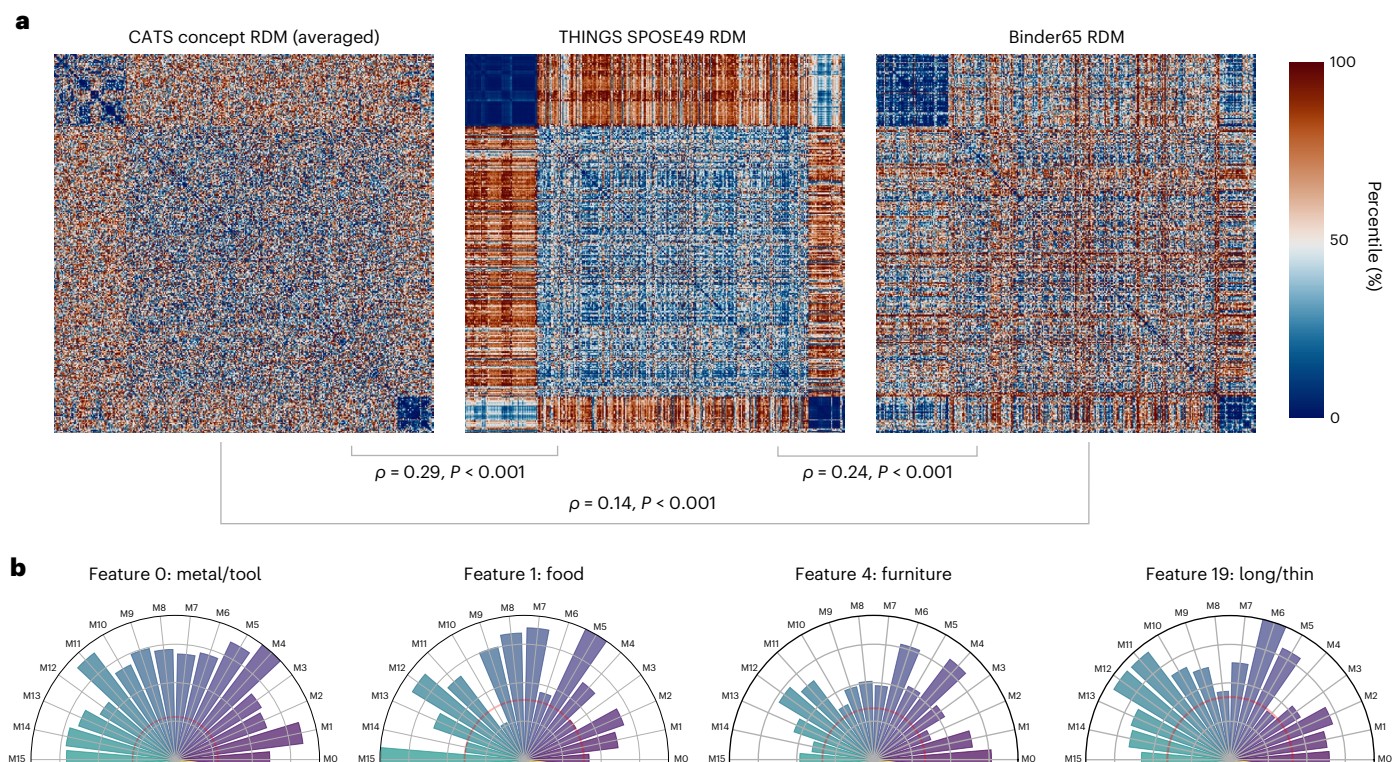

**Fig. 3 | Alignment of CATS concept layer with human semantic models.**
**a**, RDMs for the CATS concept layer, SPOSE49[9] and Binder65[8] models, computed on 332 concepts overlapping between the ImageNet-1k dataset[25] and the THINGS dataset[54] using Pearson distance. The CATS RDM represents the average across 30 independently initialized instances. Warmer colors (red) indicate greater dissimilarity between concept pairs, while cooler colors (blue) indicate greater similarity. **b**, Correlations of best-match conceptual dimensions in each CATS

Net instance with given four SPOSE49 exemplar dimensions. Each bar represents the maximum Pearson correlation between the 20 concept dimensions of a given CATS instance (represented by perimeter labels M0–M29) and the target SPOSE49 dimension (dimension names from ref. [9]). The red circle denotes the significance threshold for correlation coefficients ($r = 0.107$, two-tailed $P < 0.05$, d.f. = 330).

that semantic clustering remained consistent across layers (Fig. 4d and Supplementary Fig. 3a,b). Through RDM correlation analysis across 100 teacher–student pairs (Supplementary Fig. 3c), we found that the translation module systematically preserves semantic relationships while performing functional adaptation (Supplementary Fig. 3d for statistical significance). Specifically, RDM correlations between input and successive layers showed a gradual decrease (from 0.93 to 0.29 at the output layer, all $P < 0.001$), indicating controlled information compression rather than arbitrary loss.

**Concept transmission and evaluation (phase 3).** In the final phase, the teacher's novel concept vector (for instance, 'apple') was passed through the trained translation module, mapping it into the student's concept space. The student net was then evaluated on its ability to perform Yes/No judgments on input images using only this transferred concept vector. Across all 100 rounds of the experiment, the student networks demonstrated a remarkable ability to utilize the communicated concepts, performing consistently and significantly above the 0.5 chance level (mean accuracy (s.d.) 0.7292 (0.0781), threshold 0.5, $t(99)$ = 29.33, one-tailed $P < 0.001$, Cohen's $d = 2.93$, 95% CI [0.7137, 0.7448]).

These results validate the effectiveness of well-trained concepts in knowledge transfer. They imply that independently emerging concept

spaces across separate networks share a common lexical-semantic structure, enabling the acquisition of new knowledge without updating the high-dimensional network parameters.

## Compatibility of CATS Net with human language- and behavioral-derived concept spaces

To test whether our CATS Net is able to directly utilize the concept space generated by humans, we evaluated its performance using both language-derived and direct human behavioral-derived concept spaces.

**Word2Vec-based concept space.** We first used the Word2Vec space as the low-dimensional concept space and evaluated the ability of these human language-derived vectors to configure the TS module in CATS Net. To this end, we conducted a leave-one-out concept-abstraction experiment, which is composed of two phases (Fig. 5a). Specifically, in the first phase, the system was trained with fixed Word2Vec from 99 concepts, and learnable network parameters (both CA and TS modules), as described above. Individual category names, represented by their corresponding word vectors, were projected into 20-dimensional space and fed into the CA module. In the second phase, the remaining category (also referred to as the conceptual inferred category) was evaluated by performing Yes/No judgment given an unlearned word

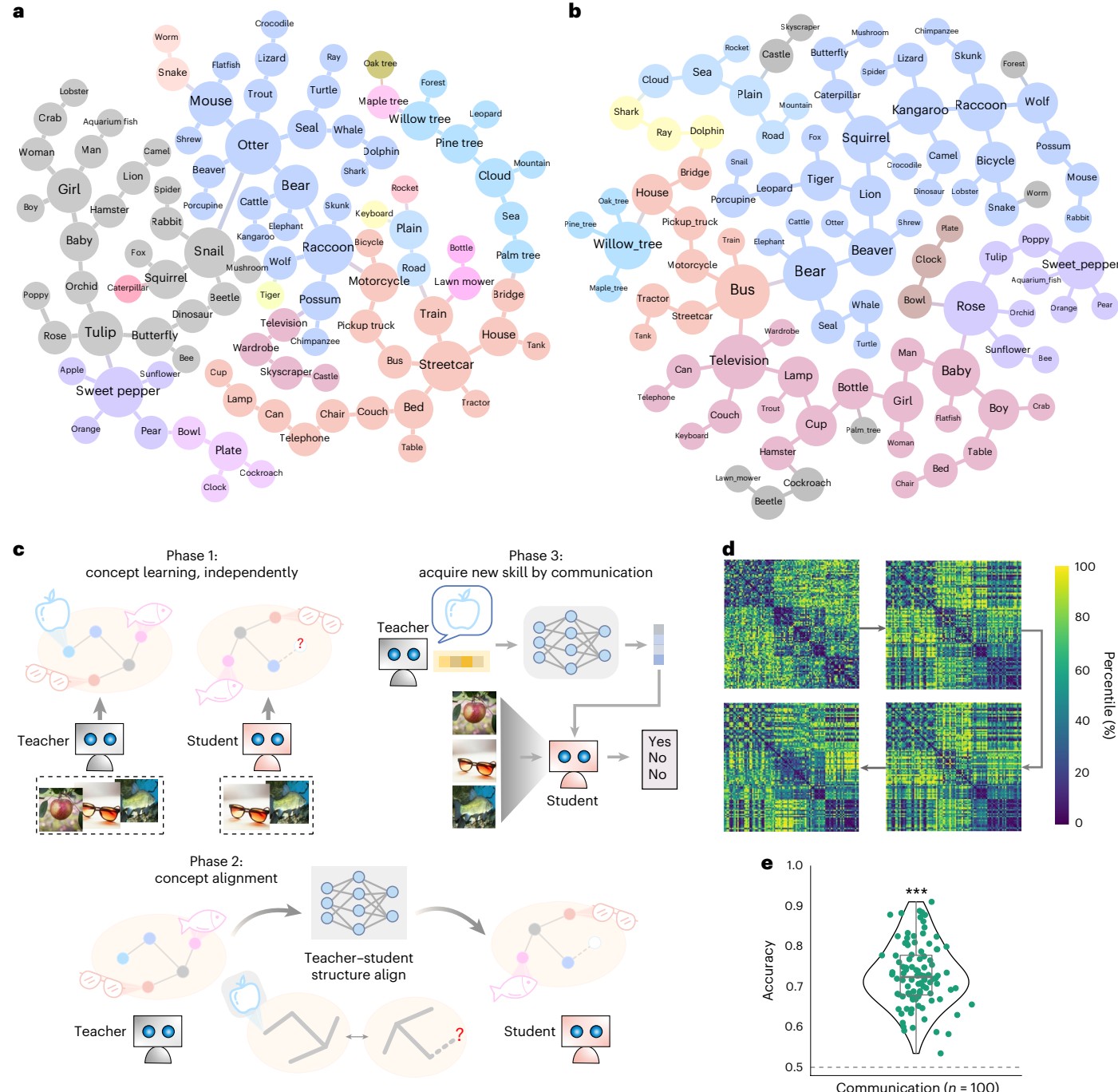

**Fig. 4 | Knowledge transfer via communication between independently trained CATS Nets. a,b**, Semantic maps of the concept space formed by the teacher (**a**) and the student (**b**) networks. Colors represent clusters at a given hierarchical clustering threshold. Manual adjustments were then made to achieve the closest possible visual alignment between the teacher net and student net clusters in the visualization. **c**, Pipeline for knowledge acquisition via communication between the teacher and student nets, consisting of three phases: independent concept abstracting, concept alignment and concept transmission. **d**, Layer-wise RDMs of the translation module (in order of arrows: input layer, third layer, seventh layer and output layer; see Supplementary Fig. 3a for a complete view of all layers). **e**, Performance of transferred concepts on CIFAR-100 for the student net through communication. Each dot represents the accuracy of an independent model instance ($n = 100$ independent experimental

units), where each was trained on a unique 99-category subset and evaluated on the corresponding held-out category. For all violin plots, individual dots represent independent model instances. The unit of analysis is a single model. Violin plots show the kernel density estimation of the data distribution. Overlaid box plots indicate the median (center line), interquartile range (25th–75th percentiles), and minimum–maximum range (whiskers). Statistical significance for accuracy was determined by a one-tailed one-sample $t$-test against the 0.5 chance level. ***$P < 0.001$. For all comparisons, the statistic values, degrees of freedom and exact $P$ values are provided in the Source data. No technical replicates were used. Unless otherwise specified, the sample violin conventions are applied in all figures. All images from PublicDomainPictures and Free-Images under a Creative Commons license CC0.

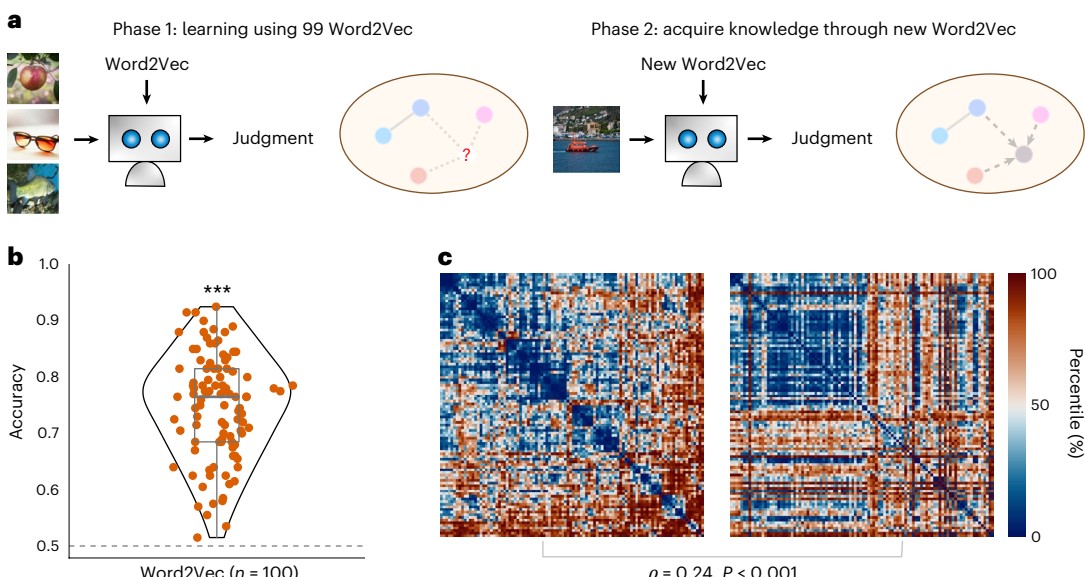

**Fig. 5 | Concept acquisition in CATS Net using Word2Vec embeddings.**
**a**, Pipeline for concept acquisition in CATS Net on CIFAR-100 using Word2Vec. In phase 1, images from 99 categories and their name embeddings (as predefined concept vectors) are used to train a randomly initialized CATS Net by updating only network parameters. In phase 2, the remaining category and its Word2Vec embedding (as an unseen concept vector) is used to evaluate the model's understanding of the novel concept. **b**, Performance on unseen concepts under the leave-one-out approach described in **a**. Each dot represents the accuracy

of an independent model instance ($n = 100$ independent experimental units), where each was trained on a unique 99-category subset and evaluated on the corresponding held-out category. Statistical significance for accuracy was determined by a one-tailed one-sample $t$-test against the 0.5 chance level. \*\*\*$P < 0.001$. **c**, RDMs of learned concept vectors (left) versus Word2Vec vectors (right). All images from PublicDomainPictures and Free-Images under a Creative Commons license CC0.

vector as the concept input. By traveling through all categories with a leave-one-out approach, we demonstrate that the accuracy of each conceptual inferred category is well beyond the chance level (mean accuracy (s.d.) 0.7474 (0.0934), threshold 0.5, $t(99) = 26.51$, one-tailed $P < 0.001$, Cohen's $d = 2.65$, 95% CI [0.7289, 0.7660]). Despite never encountering the images or category names before, the system successfully recognized the majority of images (Fig. 5b).

In addition, we assessed the similarity between the concept vectors generated by CATS Net and Word2Vec's vector representations. Although the Word2Vec vectors are derived from word co-occurrence statistics in large text corpora, which is fundamentally different from our model, their RDMs still show a significant correlation with ours (Spearman's $\rho = 0.24$, Mantel $P < 0.001$, 10,000 permutations; bootstrap 95% CI [0.154, 0.366], 5,000 resamples; Fig. 5c).

**Human behavioral data-based concept space.** To further validate our architecture's compatibility with human behavior-derived concept spaces, we conducted experiments using the SPOSE49 model. Using these human-generated concept vectors, we replicated the leave-one-out experimental approach described above. The results demonstrate that our CATS Net architecture achieves comparable performance when configured with human behavioral data (mean accuracy (s.d.) 0.6967 (0.1582), threshold 0.5, $t(99) = 12.43$, one-tailed $P < 0.001$, Cohen's $d = 1.24$, 95% CI [0.6653, 0.7281]), confirming that our dual-module framework can effectively exploit not only language-derived concept spaces but also genuine human perceptual and conceptual structures (Extended Data Fig. 2). This validation strengthens the claim that our architecture provides a general computational framework for concept formation and understanding that is compatible with authentic human cognitive processes.

These results collectively suggest that the framework proposed here can effectively exploit the information structure in humans' concept space to solve new tasks, providing a common computational framework for concept formation and understanding across totally different systems.

## Mapping neural representations of visual and semantic-control networks

To assess the extent to which the concept spaces generated by our CATS Net models align with those in the human brain, we compared similarity patterns between the model concept spaces and the visual cortex activities during object perception task using RSA. These analyses utilized our previously published functional MRI (fMRI) dataset[30], which contained human brain activity data to 95 objects that cover 3 common object domains (32 animals, 35 small manipulable artifacts and 28 large non-manipulable artifacts). Participants ($N = 26$ in the analyses) viewed images presented on the screen and performed an oral picture-naming task (Fig. 6a). Given that our models are trained solely on visual categorization tasks, we first conducted a region-of-interest (ROI)-based RSA targeting the ventral occipitotemporal cortex (VOTC) for object perception[31], defined by contrasting picture viewing versus rest (Benjamini–Hochberg false discovery rate (FDR) correction at the voxel level $q < 0.05$; see Methods for details). We computed the partial Spearman's rank correlation between the model's concept layer and human VOTC representations, explicitly controlling for low-level visual features (that is, the ResNet sensory input layer).

As shown in Fig. 6c (left), the representational patterns of concept layers of the CATS Net model showed highly significant correlation with human VOTC activity patterns (Fisher-$z$ mean (s.e.) $\rho = 0.04$ (0.004), $t(29) = 9.27$, one-tailed $P < 0.001$, Cohen's $d = 1.70$). This indicates that the abstraction mechanism in CATS Net captures conceptual representations aligning with human neural coding significantly beyond canonical visual features.

We next examined the CA module. This module dynamically gates feature representations, a function analogous to the human semantic-control network, which has been assumed to selectively access and manipulate meaningful conceptual information in relevance to a particular context or task[24]. As the semantic-control network modulates access to information rather than representing visual content itself, we assessed alignment without controlling for the sensory input layer. The first layer of the CA module (CA1) showed

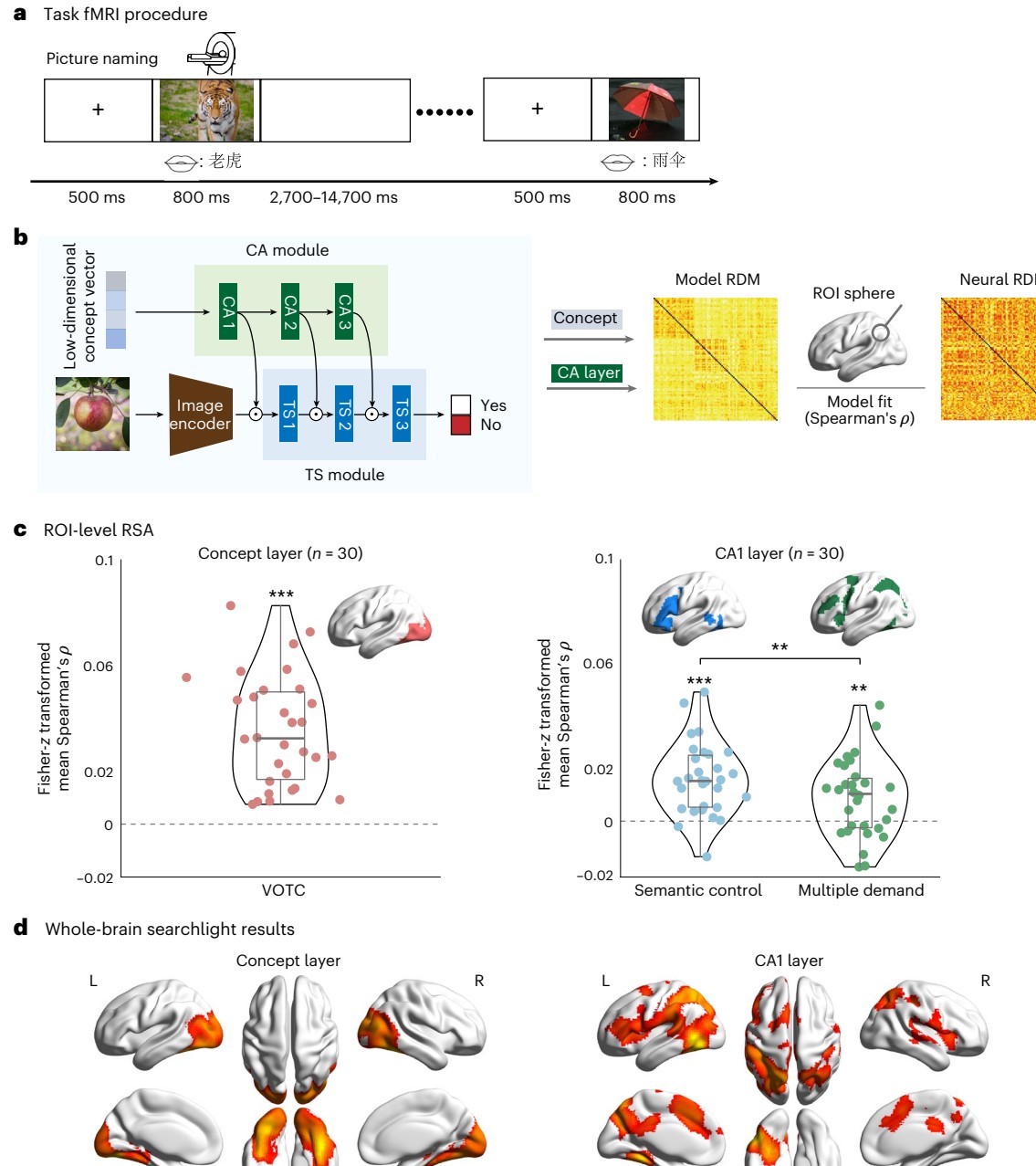

**a** Task fMRI procedure

**b**

**c** ROI-level RSA

**d** Whole-brain searchlight results

**Fig. 6 | Representational similarity between CATS Net layers and human brain.** **a**, Task fMRI experimental design for object naming. The pictures used in the fMRI task were changed to pictures under a Creative Commons license CC0. **b**, RSA pipeline. Neural RDMs were correlated with model RDMs to quantify correspondence. **c,d**, ROI-level (**c**) and whole-brain searchlight (**d**) RSA results. For both analyses, correlations were first averaged across 26 individuals for each independently trained model (*n* = 30). **c**, Correspondence between the concept layers of CATS Nets and VOTC (left), and between the CA1 layer and semantic-control[24]/multi-demand networks[32] (right). Each dot represents one model. For single-group comparisons, significance was determined by a one-tailed one-sample *t*-test against zero. Between-group differences were assessed using paired *t*-test. Asterisks indicate significance levels: **\*\*P* < 0.01; \*\*\*P* < 0.001. **d**, Group-level *t*-value maps for concept layer (left) and CA1 layer (right). Searchlight results were thresholded at voxel-level *P* < 0.001, one-tailed, and cluster-level family-wise-error-corrected *P* < 0.05. All images from Free-Images under a Creative Commons license CC0.

significant correspondence with the semantic-control network (Fisher-*z* mean (s.e.) *ρ* = 0.02 (0.003), *t*(29) = 6.44, one-tailed *P* < 0.001; additional CA module layers also showed significant correlations; Extended Data Fig. 3). Crucially, this alignment showed functional specificity: while the CA1 layer also correlated with the domain-general multiple-demand network[32] (Fisher-*z* mean (s.e.) *ρ* = 0.01 (0.003), *t*(29) = 3.22, one-tailed *P* < 0.01) (Fig. 6c, right; for other CA layers, see Extended Data Fig. 3a), the alignment with the semantic-control network was significantly stronger (paired *t*-tests: Fisher-*z* mean difference (s.e.) = 0.01 (0.003), *t*(29) = 2.89, two-tailed *P* < 0.01). These

findings collectively suggest that the CA module aligns specifically with the semantic-control processes.

Across ROIs, effect size was reliable across 30 independently initialized models (one-sample tests on Fisher-$z$ means) and should be interpreted relative to the noise ceilings (VOTC-concept $NC_z$ = 0.25; multiple-demand-CA1 $NC_z$ = 0.27; semantic-control-CA1 $NC_z$ = 0.24; Methods). For context, the VOTC correspondence of a widely used baseline model (ResNet50) and SPOSE49 model were 0.007 and 0.056, respectively (mean Fisher-$z$-transformed $\rho$, mean across 26 participants).

Finally, we confirmed these ROI-based findings using whole-brain searchlight RSA[33] (Fig. 6b). At the threshold of voxel-level $P < 0.001$, one-tailed, cluster-level family-wise-error-corrected $P < 0.05$, we found that the concept layer across all 30 CATS Nets showed significant correspondence with the bilateral VOTC (Fig. 6d, left). In contrast, the CA1 layer corresponded to regions typically associated with the semantic-control network, including the bilateral dorsomedial prefrontal cortex, inferior parietal lobe, left inferior frontal gyrus, lateral occipital complex and posterior fusiform gyrus (Fig. 6d, right; for other CA layers, see Extended Data Fig. 3b). These findings aligned with our ROI results, confirming that the concept layer corresponds to neural representations in VOTC, while the CA module predominantly corresponds to activities in semantic control regions (for validation, see Extended Data Figs. 3 and 4 and Supplementary Fig. 4).

### Emergent cross-model consensus increases alignment with human semantic systems

Our previous analyses demonstrated that model-generated concept representations showed significant model-group-level correlations with human visual cortex activity. Here we zoom into the individual model spaces to examine whether convergent patterns among independently trained CATS Nets might predict stronger alignment with human semantic systems.

Biological neural systems often show evolutionary convergence in their semantic coding strategies[34], suggesting that optimal solutions emerge under similar hardware constraints. To test whether this principle applies to our networks, we first performed cluster analysis on 30 independently trained CATS Nets. This analysis revealed a dominant organizational pattern emerging in 47% of models (14/30; Extended Data Fig. 5, left). We defined these 14 models as our 'high-consensus' group, hypothesizing that their shared representational structure might indicate greater biological plausibility compared with the remaining 16 models (the 'low-consensus' group).

To evaluate this hypothesis, we conducted two complementary analyses examining the alignment between these model groups and two independent measures of human semantic representation. First, we assessed alignment with the Binder65 neurobiological semantic model. High-consensus models showed significant better correspondence with Binder65 (mean difference (s.e.) = 0.11 (0.01), $t(29)$ = 6.98, two-tailed $P$ < 0.001; Extended Data Fig. 4), suggesting the emergent consensus structures capture neurobiologically relevant semantic dimensions. Second, we tested whether this consensus advantage extended to alignment with actual brain activity. High-consensus models also showed stronger correspondence with VOTC activity patterns than low-consensus models (mean difference (s.e.) = 0.02 (0.00), $t(29)$ = 3.05, two-tailed $P$ < 0.01; Extended Data Fig. 5, right), while controlling for the pretrained sensory input-layer RDM. These findings imply that both artificial and biological systems might be governed by equivalent optimization imperatives in how they organize semantic information. The spontaneous emergence of shared representational geometry across independently trained networks suggests that when subjected to comparable computational constraints, different systems—biological or artificial—tend to converge toward similar semantic coding solutions. This convergence may reflect fundamental organizational principles that efficiently support semantic processing across different types of intelligent system.

## Discussion

The CATS Net architecture offers a unified computational framework for linking raw sensory experience with symbolic thought. By integrating concept formation through compression and understanding through sensorimotor reinstatement, this dual-module system provides a computational account of how high-dimensional sensory inputs are mapped into low-dimensional, communicable conceptual spaces. This process moves beyond the limitations of purely language-derived representations by demonstrating that functionally useful conceptual structures can emerge directly from task-driven sensorimotor grounding[35]. Such a mechanism is particularly vital for capturing nuanced or domain-specific knowledge that is often difficult to articulate fully through natural language—a form of intuitive, embodied expertise that can now be computationally instantiated and utilized within artificial neural networks.

The implications of this grounded representational mechanism also suggest a different perspective on emergent communication. While many current approaches rely on end-to-end joint optimization via backpropagation[36–38], CATS Net motivates a more modular and biologically plausible alternative: agents can develop internal conceptual structures independently and subsequently align them through a shared symbolic interface. This approach captures a foundational principle of human interaction, where low-dimensional symbols are used to reactivate rich, high-dimensional sensorimotor experiences in others. In principle, such a design may reduce the dependence on continual, large-scale joint re-optimization and facilitate incremental alignment in decentralized multi-agent settings, providing a plausible route toward scalable collective intelligence.

Beyond its implications for artificial intelligence, the framework offers a concrete hypothesis that connects to established neurocognitive accounts. The observed correspondence between CATS representations and brain activity patterns in human VOTC and semantic control network is consistent with theories proposing top-down modulation of feature representations during semantic processing[5]. In this view, task-related feature gating—implemented here as a multiplicative, element-wise interaction—serves as a candidate mechanism through which relevant semantic features are amplified and irrelevant ones suppressed according to shifting task demands. While other forms of modulation exist, formalizing this multiplicative account helps translate descriptive theories into testable computational predictions about the neural basis of semantic flexibility.

However, the current scope of CATS Net is primarily constrained to concrete concepts with identifiable visual referents. Abstract concepts (for instance, 'justice' or 'freedom') pose additional challenges due to their lack of bounded physical referents and high inter-individual variability[39,40]. Although our exploratory analyses suggest that CATS Net's concept layers capture representations related to abstract dimensions, the model was not explicitly optimized for the fuzzy boundaries and limited annotation reliability characteristic of non-perceptual categories[41]. A key direction for future work is therefore to extend this architecture to multimodal integration, examining how the joint constraints of vision, audition and language can further sharpen conceptual boundaries and support more abstract representational spaces. More broadly, systems that combine conceptual compression with sensorimotor reinstatement may offer a practical step toward artificial intelligence that represents—and communicates about—the world in a more human-like, grounded manner.

## Methods

### Hierarchical gating of CATS Net

In a concept-abstraction task on an image dataset, we use a 20-dimension real-valued vector to present each category. This dimension was selected from a tested range of 10, 20, 100, as it offered the optimal trade-off between compression efficiency and representational capacity (Supplementary Fig. 1a). The compactness of this vector

space, compared with the high-dimensional parameter space of the neural network, reflects the highly compressed nature of the concepts. The model's pipeline begins with a pretrained ResNet50 backbone[16], chosen over Vision Transformer[17] for its computational efficiency after observing similar performance from both (we use official IMAGE-NET1K_V1 weights from https://pytorch.org/ for both backbones). The extracted 2,048-dimension features are then fed into the TS module. This module is a 3-layer perceptron ([2,048-100-100-2]) with batch normalization and rectified linear unit (ReLU) activation. The 3-layer architecture was adopted for its demonstrated robustness, as our tests with 1, 3 and 5 layers all yielded comparable performance (Supplementary Fig. 1a). To match this structure, the CA module is also a 3-layer perceptron ([20-2,048-100-100]), which takes the 20-dimension concept vector as input and uses the Sigmoid function $\sigma(x) = \frac{1}{1+e^{-x}}$ to generate controlling signals between 0 and 1. The output layer of the TS module consists of two neurons for 'Yes' (0, 1) and 'No' (1, 0) classification, optimized using a cross-entropy loss.

Consider a multi-layer perception of $L+1$ layers, indexed by $l = 0, \cdots, L$ with $l = 0$ and $l = L$ being the input and output layers. Let $W_l^{\text{TS}}$ be the connection weight between the $(l-1)$th layer and $l$th layer in the TS module, and $W_l^{\text{CA}}$ for the CA module. $\mathbf{x}_{l-1}$ denotes the input of the connections $W_l^{\text{TS}}$ and $\mathbf{c}_{l-1}$ denotes the input of the connections $W_l^{\text{CA}}$. $\mathbf{o}_l$ denotes the output of the connections $W_l^{\text{TS}}$ and $\mathbf{g}_l$ denotes the output of the connections $W_l^{\text{CA}}$. It is clear that the dimension of feature extractor output is the same with the $\mathbf{x}_0$ and the dimension of concept vector is with $\mathbf{c}_0$. For the CA module, after applying normalization and activation to the $\mathbf{g}_l$, it will be the input $\mathbf{c}_l$ for weight $W_{l+1}^{\text{CA}}$.

The CA module does not need to directly modulate the feature extractor, because it is possible to utilize gating signals in a much easier way to modulate the processing of the feature extractor. The dimensions of $\mathbf{x}_{l-1} \in \mathbb{R}^d$ and $\mathbf{g}_l \in \mathbb{R}^d$ are consistent from $l = 1$ to $L$. For $l = 1$ to $L$, let $\mathbf{z}_{l-1} = \mathbf{x}_{l-1} \odot \mathbf{g}_l$, $\mathbf{z}_{l-1} \in \mathbb{R}^d$, we replace the $\mathbf{x}_{l-1}$ by $\mathbf{z}_{l-1}$ and set it as input for weight $W_l^{\text{TS}}$. Operator $\odot$ is the Hadamard product (also known as the element-wise product), is a binary operation that takes in 2 matrices of the same dimensions and returns a matrix of the multiplied corresponding elements. In our case, the hierarchical gating take place at [2,048-100-100] layers between the CA and TS modules.

Under this network structure, even if the same stimulus $\mathbf{x}$ is provided to the TS module, CATS Net will conduct hierarchical gating operations based on the different concept vector given to the CA module. Let $H(\mathbf{x}, \mathbf{c})$ be CATS Net, so $H(\mathbf{x}, \mathbf{c}) = G(T(\mathbf{x}), C(\mathbf{c}))$ where $T(\cdot)$ is the TS module, $C(\cdot)$ is the CA module and $G$ is the hierarchical gating between CA module and TS module. When such gating only takes effect at a certain layer, it is equivalent to scaling the data of the current layer, and when it acts on multiple layers along with activation functions, provided that the input stimulus $\mathbf{x}$ remains unchanged, there exists a variation of $T'$ such that $H(\mathbf{x}, \mathbf{c}) = T'(\mathbf{c})$. That is to say, it is equivalent to realizing several different TS parameters with distinct concept vectors.

### Concept-abstraction task data and training

The input to CATS Net includes concept vector and natural images while the output is a two-dimension vector indicating 'Yes' or 'No'. Therefore, the original image–label doublet vision dataset will be converted to the image–concept–label triplet one. Taking the ImageNet-1k dataset used in the current work as an illustration, we randomly sampled 1,000 points in a 20-dimension real vector space and assigned them to each category, which was fed to the CA module as the initial for the abstracted concept. Then for half of images in the whole dataset, we assigned the corresponding concept vector to them, along with the 'Yes' label. For the other half, we randomly assigned a non-corresponding concept vector with label of 'No', as negative samples for training stability.

Two training phases first begin with the network-learning phase: the concept vector inputs to the CA module were fixed, while all network parameters, including those in both CA and TS modules, were updated by gradient back-propagation using a binary supervising

signal (Yes/No), to indicate whether the image belongs to the target concept category or not. In the following concept-learning phase, only concept vectors were modified by the back-propagated gradients, with all network parameters fixed. The two phases in the training process were carried out alternately in an epoch-by-epoch manner. This provides better interpretability of concept-learning dynamics, which implies the learning of concept space can be independent from the learning of network parameters. We also validated this approach against end-to-end joint training and found comparable performance (Supplementary Fig. 1a), confirming that the concept-formation process is robust to training methodology. The training was terminated after 5 epochs to ensure accuracy reaching the plateau. Uniform distributed noise ranging from −0.1 to 0.1 was injected into each element of the concept vectors, in both the network-learning phase and the concept-learning phase. We found that it effectively increased the system's robustness for distinguishing various categories. No noise was added to the concept vectors in the testing. In all experiments, the length of concept vectors was set to 20, that is, they contained 20 real elements.

### Visualization of configured CATS Net by CAM

We made a little modification for traditional Grad-CAM[26], to directly show the importance of each neuron in the last layer of the pretrained feature extractor, after being gated by the signals from CA module. To obtain the class-discriminative localization map $L_{\text{Grad-CAM}}^{\mathbf{c}} \in \mathbb{R}^{u \times v}$ of width $u$ and height $v$ for any concept $\mathbf{c}$, we first compute the gradient of the 'Yes' score $y^{\mathbf{c}}$ (the value of 'Yes' neuron before the softmax, given concept input), with respect to feature maps $A^k$ of a convolutional layer, that is, $\frac{\partial y^{\mathbf{c}}}{\partial A^k}$. These gradients flowing back are global average-pooled to obtain the neuron importance weights $\alpha_k^{\mathbf{c}}$

$$\alpha_k^{\mathbf{c}} = g_{1,k} \frac{1}{Z} \sum_i \sum_j \frac{\partial y^{\mathbf{c}}}{\partial A_{ij}^k}$$

where $i$ and $j$ are the index for width $u$ and height $v$, that is, the pixel in each two-dimensional convolution kernel, and $Z$ is the total number of pixels in this kernel. $k$ stands for the index of convolution kernel; it is straightforward that the number of convolution kernels is the same as the dimension of the gating vector $\mathbf{g}_1$. The $g_{1,k}$ is the $k$th element of $\mathbf{g}_1$, that is, gating signals from the output of $W_1^{\text{CA}}$.

This weight $\alpha_k^{\mathbf{c}}$ represents a partial linearization of the deep network downstream from $A$, and captures the 'importance' of feature map $k$ for a target concept $c$. We perform a weighted combination of forward activation maps and follow it by a ReLU to obtain

$$L_{\text{Grad-CAM}}^{\mathbf{c}} = \text{ReLU}\left(\sum_k \alpha_k^{\mathbf{c}} A^k\right)$$

Finally, $L_{\text{Grad-CAM}}^{\mathbf{c}}$ is linearly scaled to the size of the input image so as to obtain the activation map shown in Fig. 2b.

### Hyper-category functional specificity of the basis vector of concept space

We assigned a hyper-category label to each category in ImageNet-1k, using WordNet from the nltk library. Specifically, as each class label in ImageNet has its corresponding synset ID in WordNet[42], we first obtain all the hyper synsets of each class in WordNet to form a WordNet synset chain for that class. Subsequently, we examine the synsets one by one from the top of the synset chain downward to check whether they correspond to the four preset hyper-categories such as mammals and artifacts. If none of them can be matched, then the hyper-category label of 'others entity' will be assigned to that class. The synset tokens for four hyper-categories were 'mammal.n.01', 'animal.n.01', 'instrumentality.n.03' and 'artifact.n.01'. Thus, the five hyper-categories were 'mammal', 'others mammal', 'instrumentality', 'others artifact' and 'others entity'.

For a well-trained CATS Net, given the one-hot vectors ranging from dimension 1 to 20, we calculated the number of images with a 'Yes' response over the evaluation set of ImageNet-1k (50,000 images), for each hyper-category.

### Functional entropy

For each concept vector in the concept space, we define the functional entropy as

$$e = -\sum_i p_i \log p_i$$

where

$$p_i = \frac{c_i}{\sum_j c_j}$$

The $c_i$ stands for the number of 'Yes' response to $i$th class across the whole classes in the dataset and the $p_i$ is the normalized probability prepared for entropy calculation. A higher value of functional entropy also implies that the current input concept vector has a relatively even selectivity for each category. In other words, this vector cannot represent the concept of a specific category in the dataset. On the contrary, a lower entropy indicates that the concept vector is more inclined to respond 'Yes' to certain specific categories while answering 'No' for the majority of the remaining categories. So the distribution of the functional entropy reflects the overall attribution over the whole concept space.

### Hierarchical clustering analysis of CIFAR-100 concept set

We used hierarchical clustering (Matlab function dendrogram) to group concept vectors generated by CATS Net, based on cosine distance between vectors and unweighted average linkage between clusters. Specifically, two concepts, each from one of two distinct but connected branches or leaves in the dendrogram with the closest distance, were connected by one edge. We traversed all pairs of connected branches and leaves, linking all pairs of concept nodes to meet the requirement of the closest distance. The visualization of the semantic network was generated by Gephi[43].

### Leave-one-out training and concept vector expansion

This section describes the technical procedures underlying the communication experiment presented in Fig. 4. The leave-one-out training and concept vector expansion serve two critical functions: (1) creating knowledge asymmetry between teacher and student networks, and (2) generating sufficient training data for the translation module that enables concept transfer.

For the leave-one-out training, the student CATS Net was trained on dataset $D_{99}$ containing images and labels from 99 categories, while one category $D_1$ was withheld to create the knowledge gap that communication aims to bridge. The teacher net was trained on the complete dataset including $D_1$.

Subsequently, to generate expanded concept vectors for training the translation module, concept vector expansion for the withheld category $D_1$ was performed through concept manipulation only, that is, only through the concept-abstraction phase without retraining the network parameters. To utilize the concept obtained so far to identify $D_1$ as much as possible, we introduced a repelling loss $L_{rep}$ for learning a new concept, which was defined as

$$L_{rep}(C, C_{old}, \tau) = \sum_{C_i \in C_{old}} \exp(-|C_i - C|^2/\tau)$$

where $C_i \in C_{old}$ are the concepts of categories belonging to $D_{99}$ and $C$ is the concept of the remaining category in $D_1$. To test the system's capability of few shot learning, only 2 images belonging to $D_1$ and 1 image from each of the 99 learned categories belonging to $D_{99}$ were utilized in concept abstraction. The concepts assigned to the category in $D_1$ were randomly initialized and trained to minimize the following loss function

$$L = L_{CE}(x_{new}, y|C) + \alpha L_{CE}(x_{old}, \bar{y}|C) + \beta L_{rep}(C, C_{old}, \tau)$$

where $L_{CE}$ denotes the cross-entropy loss, $x_{new}$ is the image sampled from the new category in $D_1$, $x_{old}$ is the image sample from the learned categories in $D_{99}$, $y$ is the label 'Yes', $\bar{y}$ is the label 'No', and $\alpha$ and $\beta$ are parameters used to balance the different contributions of the losses. Hyperparameters in these experiments were set to $\alpha = 0.5$, $\beta = 0.001$, $\tau = 0.01$ and the learning rate lr $= 0.01$.

### Data expansion and translation module

Building on the leave-one-out training procedure described above, this section details the data expansion process and translation module training that enable concept transfer between teacher and student networks. The datasets used in learning-by-communication experiment was CIFAR-100. The teacher net was trained with dataset D of all 100 categories, while the student Net was trained with $D_{99}$ containing 99 categories. An additional translation module was then trained to map the concept from teacher net to student net. First, according to the procedure for training CATS Net, the teacher net generated one concept for each category in D $(D = D_{99} \cup D_1)$, and the student net generated one concept for each category in $D_{99}$. To generate enough samples for training this map, the teacher concept dataset was extended to 97 concept vectors for each category by concept vector expansion described above. Specifically, after the initial training of the teacher net, the network parameters were fixed. Then 96 additional different concept vectors for each category were obtained through the training procedure described in the 'Leave-one-out training and concept vector expansion' section.

The translation module used was a multiple-layer perceptron, with 10 hidden layers containing 500 neurons each. The ReLU activation function and the mean-squared-error loss function were used. During the training, the dropout probability of all hidden layers was set to 0.3. The translation module was trained, for $D_{99}$, to map the 97 concept for each category from the teacher net to the corresponding 1 concept for each category from the student net. The learning rate was set to 0.0001 and decayed by a factor of 0.5 for every 10 epochs. The Adam[44] algorithm was used. The training lasted for 200 epochs to ensure convergence. The experiment was repeated 100 rounds, with a different class chosen as $D_1$ in each round.

### Semantic detail preservation analysis

To assess whether the translation module preserves semantic details during concept transfer, we conducted layer-wise representational analysis across all 100 teacher–student pairs. For each translation module, we extracted feature vectors from the input layer, all 11 ReLU hidden layers and the output layer when processing the teacher's 100 concept vectors. These 13-layer feature representations were analyzed using RDM correlation analysis for quantitative assessment of information preservation. For RDM analysis, we computed pairwise Euclidean distances between all concept representations within each layer, then calculated Spearman rank correlations between layer-wise RDMs. Statistical significance was assessed using two-tailed $t$-tests across the 100 translation modules. This analysis revealed systematic preservation of semantic relationships throughout the translation process, with gradual but controlled information compression across layers.

### Word2Vec as concept

In these experiments, CATS Net was trained using the category-name word vectors as the predefined concept vector, which were provided by the fastText library[45]. We used the pretrained 300-dimensional English word vectors (cc.en.300.bin) and reduced them to 20 dimensions using fastText's built-in `reduce_model()` function, which uses principal component analysis for dimensionality reduction. This 20-dimensional representation was chosen to match the dimensionality of our learned concept vectors, enabling direct comparison between the two concept

spaces. The dataset was divided into two parts, $D_{99}$ and $D_1$, in the same way as in the leave-one-out concept-abstraction experiment. CATS Net was directly trained by class label names, represented by their 20-dimensional Word2Vec embeddings, with images belonging to $D_{99}$. Then it was tested with the untrained class name corresponding to $D_1$ to identify the images. Experiments were also repeated 100 rounds with each category chosen as $D_1$.

#### THINGS SPOSE49 and Binder65 as concept

First, we identified 334 shared concepts between the ImageNet-1k and THINGS datasets. Both the ImageNet-1k and THINGS datasets provide category labels with unique synset ID in WordNet[42]. By matching these IDs, we extracted 334 shared concepts. Feature vectors for these concepts were then extracted from the SPOSE49 model provided in ref. 9. For Binder65, feature vectors for each object name were computed as Pearson's correlation coefficients between the object name embeddings and the Binder65 dimension name embeddings in the Word2Vec embedding space[46]. Two concepts could not be represented in the Binder65 feature space, resulting in a final set of 332 concepts for subsequent analyses. RDMs were constructed using pairwise Pearson's distance (that is, 1 − Pearson's correlation coefficient) between feature vectors.

For analyses focusing on specific Binder65 subdomains, RDMs were computed using the corresponding subset of Binder65 dimensions. The 'cognition' domain was excluded from these analyses owing to its single-dimensional structure, which precluded the calculation of meaningful dissimilarity matrices required for our analytical approach.

For the WT95 dataset, we identified 89 shared concepts between the WT95 stimulus set and the THINGS dataset. All subsequent analyses on this dataset were conducted using these 89 concepts.

#### fMRI dataset

**Participants.** Twenty-nine participants (19 female; median age, 20 years; range, 18–32 years) were recruited in our study and were scanned in a conditional-rich event-related fMRI experiment. All participants were right-handed, native Mandarin speakers with normal or corrected-to-normal vision, and had no history of neurological or language disorders. All protocols and procedures of the current study were approved by the State Key Laboratory of Cognitive Neuroscience and Learning at Beijing Normal University (ICBIR_A_0040_008). Before participation, all participants provided written informed consent. The study was conducted in accordance with the Declaration of Helsinki and adhered to all relevant ethical guidelines.

**Stimulus and procedures.** Ninety-five objects were chosen, including 3 common domains (32 animals, 35 small manipulable artifacts and 28 large non-manipulable artifacts). Each object was presented as a 400 × 400 pixels colored image displaying a representative exemplar against a white background (10.55° × 10.55° visual angle). The stimulus described above is hereafter referred to as the WT95 object image dataset. All the participants were asked to name each displayed picture using oral language. The whole experiment included six runs, with each item repeated for six times across the experiment. Each run (8 min 45 s) consisted of 95 trials, with each item presented once per run. The trial structure consisted of a 0.5 s fixation, followed by a 0.8 s stimulus presentation and an inter-trial interval ranging from 2.7 s to 14.7 s. The order of stimuli and inter-trial interval durations were randomized using the optseq2 optimization algorithm (http://surfer.nmr.mgh.harvard.edu/optseq/)[47]. Each run began and ended with a 10 s fixation period.

**Image acquisition.** Functional and anatomical MRI images were collected at the MRI center, Beijing Normal University using a 3 Tesla Siemens Trio Tim Scanner. A high-resolution three-dimensional structural image was collected with a three-dimensional magnetization prepared-rapid gradient echo (3D-MPRAGE) sequence in the sagittal plane (144 slices, repetition time 2,530 ms, echo time 3.39 ms, flip angle 7°, matrix size 256 × 256, voxel size 1.33 × 1 × 1.33 mm). Functional images were acquired with an echo-planar imaging sequence (33 axial slices, repetition time 2,000 ms, echo time 30 ms, flip angle 90°, matrix size 64 × 64, voxel size 3 × 3 × 3.5 mm with a gap of 0.7 mm).

#### WT95 RDM of CATS Net model

We have previously trained 30 different CATS Nets on ImageNet-1k and these models have distinct conceptual spaces. On the basis of these spaces, we abstracted the concepts from WT95 object image dataset to form the WT95 RDMs for each model. Specifically, for each CATS Net, we retained all the network parameter modules (TS module and CA module), discarded the concept set and then allocated 95 random initial points in 20-dimension space as 95 concept vectors. Subsequently, with the network parameters fixed, we updated only the concept vectors through the backpropagation algorithm until convergence was achieved. The dissimilarity between each pair of concepts was then calculated as 1 − Pearson's correlation coefficient to generate the 95 × 95 RDM.

To obtain a more accurate estimation of the concept space of the model through the RDM, we repeated the above concept-formation process 100 times for one model. Then, we averaged the RDMs of these 100 sets of concepts to represent the RDM of that model.

#### Preprocessing for task fMRI data

The functional images were preprocessed and analyzed using Statistical Parametric Mapping (SPM12; http://www.fil.ion.ucl.ac.uk/spm). For each participant, the first five volumes of each run were discarded for signal equilibrium. Then the remaining images were corrected for time slicing and head motion and then spatially normalized to Montreal Neurological Institute space via unified segmentation (resampling into 3 × 3 × 3 mm³ voxel size). Three individuals were excluded from the data analyses owing to the successive head motions (>3 mm or 3°). For the functional images of each participant, the object-relevant beta weights were obtained using general linear model. The general linear model contained onset regressor for each of 95 items, 6 regressors of no interest corresponding to the 6 head motion parameters, and a constant regressor for each run. Each item-relevant regressor was convolved with a canonical hemodynamic response function, and a high-pass filter cut-off was set as 128 s. The resulting *t*-maps for each item versus baseline were used to create neural RDMs.

#### ROI definition

The VOTC mask was defined as regions showing stronger activation to all pictures relative to the rest in the fMRI dataset (FDR $q < 0.05$) within the cerebral mask combining the posterior and temporooccipital divisions of inferior temporal gyrus (15#, 16#), the inferior division of lateral occipital cortex (23#), the posterior division of parahippocampal gyrus (35#), the lingual gyrus (36#), the posterior division of temporal fusiform cortex (38#), the temporal occipital fusiform cortex (39#), the occipital fusiform gyrus (40#), the supracalcarine cortex (47#) and the occipital pole (48#) in the Harvard-Oxford Atlas (probability >0.2).

#### Representation similarity analysis

For the ROI-level analysis, activity patterns for each item within each ROI were extracted from whole-brain *t*-statistic images. Neural RDMs were then generated based on the Pearson distance between activation patterns for each object pair. Model fitting was quantified by computing the partial Spearman's rank correlation (Spearman's $\rho$) between the neural RDMs and model RDMs, controlling for RDMs derived from the feature extraction layer. For the analysis of the CA module specifically, this partialling out procedure was not applied. The resulting correlation coefficients underwent Fisher-*z* transformation and were averaged at the individual level. At the model-group level, one-sample *t*-tests were

performed on the individual-level mean correlation coefficients ($\rho$ values) to determine significant differences from zero. For comparative analyses between model groups within VOTC, two-sample $t$-tests were used to evaluate differences in individual-level mean $\rho$ values. In addition, paired $t$-tests were utilized to assess statistical differences between ROIs, enabling direct comparison of regional effects across the predefined functional and anatomical boundaries.

For the whole-brain analysis, a searchlight approach was implemented wherein multivariate activation patterns within a sphere (radius 10 mm) centered on each voxel were extracted to compute Pearson-based neural RDMs. For each searchlight position, the Spearman's rank correlation (Spearman's $\rho$) between the neural RDM and model-derived RDMs was computed, partialling out the effects from the sensory input layer. For the analysis of the CA module specifically, this partialling out procedure was not applied. This procedure generated correlation maps for each participant by iteratively moving the searchlight center throughout the whole brain. The resulting correlation maps underwent Fisher-$z$ transformation and were spatially smoothed using a 6 mm full-width at half-maximum Gaussian kernel. These processed maps were then averaged across individuals to produce group-level representation. Statistical significance was assessed through model-group-level analysis using one-sample $t$-tests against zero to identify brain regions showing significant correlations with the theoretical models.

### Noise ceiling estimation
Our primary effect size for each model instance $i$ is computed by correlating the instance's RDM with each participant's RDM within a ROI using Spearman's $\rho$, applying a Fisher-$z$ transform and averaging across participants. Therefore, to provide an upper bound that is commensurate with this statistic, we estimate a noise ceiling (NC) in the $z$-domain that jointly reflects measurement reliability on the participant side and stochastic variability on the model-instance side.

For the subject-level reliability of each participant $s$, $\text{rel}_s$, we first estimate the group-mean RDM reliability $\text{rel}_{\text{group}}$ via participant split-half half-sample means correlated across many random 50/50 splits, Fisher-$z$ averaged and Spearman–Brown corrected), then relate each participant to the leave-one-out group mean, $r(X_s, \overline{X_{-s}})$, and obtain:

$$\text{rel}_s \geq \frac{r^2(X_s, \overline{X_{-s}})}{\text{rel}_{\text{group}}}$$

For single-instance model reliability, $\text{rel}_{\text{model}}$, we estimate the reliability of a single model instance using a leave-one-out approximation: for each instance $i$, we compute the correlation between $M_i$, and the mean RDM of the remaining $M-1$ instances, $r_i = \rho(M_i, \overline{M_{-i}})$; we then average $r_i$ in the Fisher-$z$-domain and back-transform to obtain $\text{rel}_{\text{model}}$.

Finally, the expected correlation between a single participant and a single instance is bounded by $\sqrt{\text{rel}_s \text{rel}_{\text{model}}}$. Because our effect averages Fisher-$z$ values across participants, we aggregate the bound in the same domain:

$$\text{NC}_z = \frac{1}{S} \sum_{s=1}^{S} \text{atanh}\left(\sqrt{\text{rel}_s \text{rel}_{\text{model}}}\right)$$

### Brain visualization
The brain results were projected onto the Montreal Neurological Institute brain surface for visualization using BrainNet Viewer[48] (version 1.7; https://www.nitrc.org/projects/bnv/; RRID: SCR_009446) with the default 'interpolated' mapping algorithm, unless stated explicitly otherwise.

### Model RDM clustering
$K$-means clustering was performed using the kmeans function in Matlab R2021a with default parameters ($k = 2$).

### Statistics and reproducibility
No statistical methods were used to pre-determine sample sizes, but our sample size ($N = 26$) is similar to those reported in previous publications investigating semantic representations (for instance, refs. 49–51).

Out of the 29 participants recruited, data from 3 individuals were excluded from the final analyses because of excessive head motion (>3 mm or 3°). No other data were excluded.

The experiments were not randomized as there were no group allocations involved in this study. The investigators were not blinded to allocation during experiments.

Data distribution was assumed to be normal but this was not formally tested. The data distributions and individual data points were all plotted.

### Reporting summary
Further information on research design is available in the Nature Portfolio Reporting Summary linked to this article.

### Data availability
Source data for Figs. 2–6 and Extended Data Figs. 1–5 are provided with this paper. The fMRI data that support the findings of this study have been deposited in the Open Science Framework (OSF) at https://doi.org/10.17605/OSF.IO/5Y8P6 (ref. 52). The embeddings of SPOSE49 model[9] are available via OSF at https://osf.io/f5rn6/files/8yjh5. In addition, the anchor word embeddings used for the Binder65 model[8] can be accessed at https://www.neuro.mcw.edu/index.php/resources/brain-based-semantic-representations/. We only used the ImageNet-1k[25] training part and validation part for CATS Net training and testing in this work. Website: https://image-net.org/index.php. Website for CIFAR-100 dataset[29]: https://www.cs.toronto.edu/kriz/cifar.html. Source data are provided with this paper.

### Code availability
The code source of all results shown in this paper is available via Zenodo at https://doi.org/10.5281/zenodo.18136642 (ref. 53) and GitHub (https://github.com/Hiroid/CATS_Net).

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

## Acknowledgements

This research was supported by grants from the CAS Project for Young Scientists in Basic Research (grant number YSBR-041 to Y.C.); the National Natural Science Foundation of China (grant numbers 32595490, 32595491 to Y.B.); Strategic Priority Research Program of the Chinese Academy of Sciences (CAS) (grant number XDB1010302 to S.Y.); the International Partnership Program of Chinese Academy of Sciences (grant number 104GJHZ2025032FN to Y.C.); the STI2030-Major Project 2021ZD0204100 (grant number 2021ZD0204104 to Y.B.); the National Natural Science Foundation of China (grant numbers 31925020 and 82021004 to Y.B.). The funders had no role in the study design, data collection and analysis, decision to publish or preparation of the manuscript. We thank D. Nikolić, F. Alexandre, J. Zhang, G. Ma, X. Wang and H. Yang for their valuable comments on earlier drafts of the paper.

## Author contributions

L.G. and H.C. conceived of the study under supervision of Y.C., Y.B. and S.Y. L.G., H.C. and Y.C. designed the experiment. L.G. and Y.C. implemented and conducted the experiments on CATS Net models. H.C. and L.G analyzed the models and fMRI data and plotted the results. L.G., H.C. and Y.C. wrote the initial draft. All authors discussed the results and reviewed and edited the paper.

## Competing interests

The Institute of Automation, Chinese Academy of Sciences holds a granted Chinese patent (Patent No. ZL 2023 1 0103748.3) covering the concept generation process of the CATS Net described in this paper (invented by Y.C. and S.Y.).

## Additional information

**Extended data** is available for this paper at https://doi.org/10.1038/s43588-026-00956-4.

**Correspondence and requests for materials** should be addressed to Yang Chen, Yanchao Bi or Shan Yu.

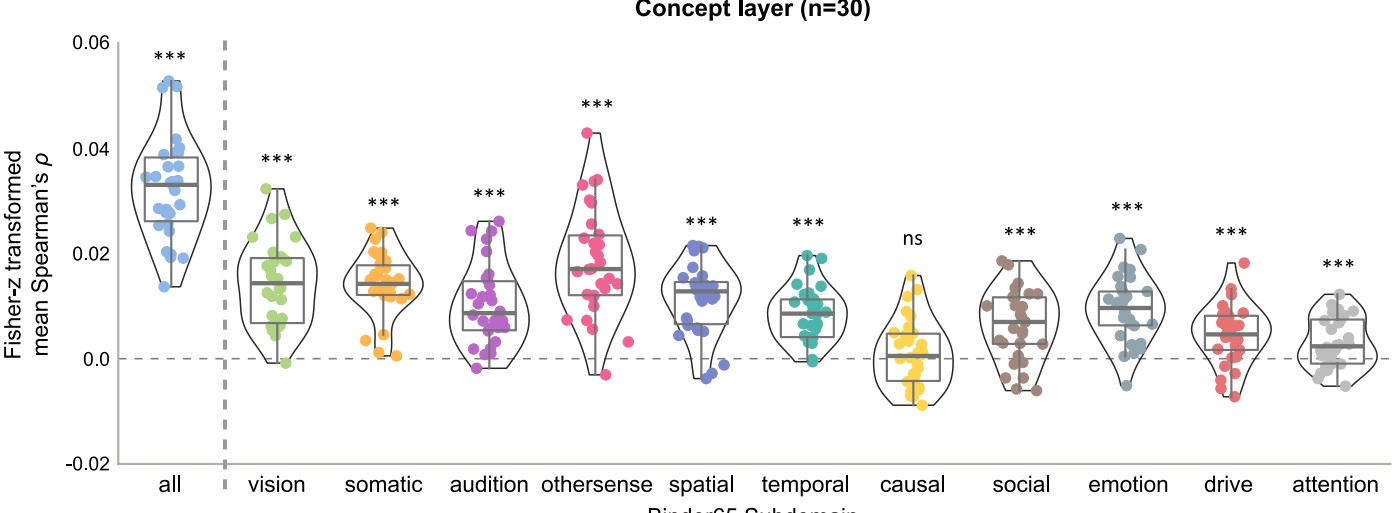

**Extended Data Fig. 1 | Representational similarity between CATS concept layer and Binder65 subdomains on ImageNet dataset.** This figure illustrates the RSA results between our CATS model's concept layer and the 11 subdomains of the Binder65 model. RDMs were generated for both the CATS concept layer and each Binder65 subdomain using WT95 stimulus dataset. The y-axis displays Fisher's z-transformed Spearman's rank correlation coefficients ($\rho$) between the respective RDMs. Individual data points represent correlation values from each independently trained model. Asterisks (***) above each subdomain indicate statistical significance ($p < 0.001$) from one-sample t-tests conducted at the group level. The "ns" indicate $p > 0.05$.

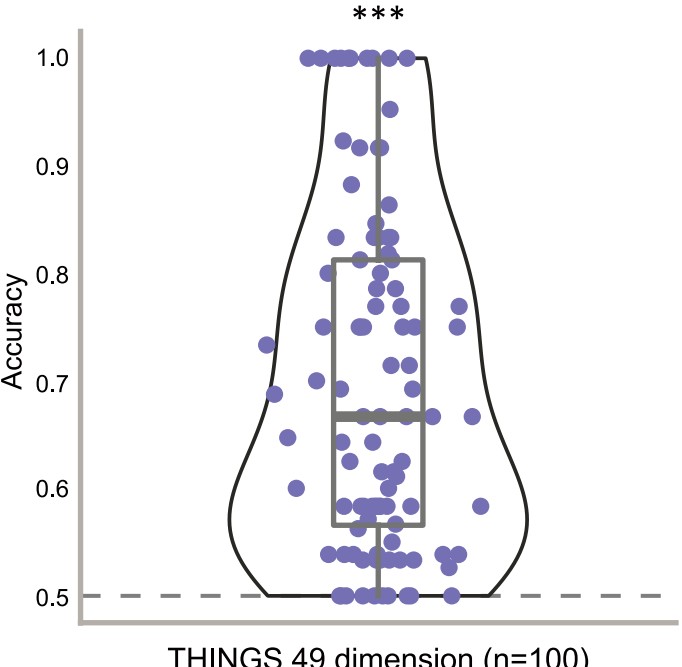

**Extended Data Fig. 2 | Performance on unseen concepts under the leave-one-out approach using THINGS and 49-dimension human-generated embedding vectors.** We randomly chose 100 categories from 1,854 object categories from THINGS as 100 independent runs of leave-one-out experiment. Each point represents the accuracy of one category chosen for leave-one-out experiment.

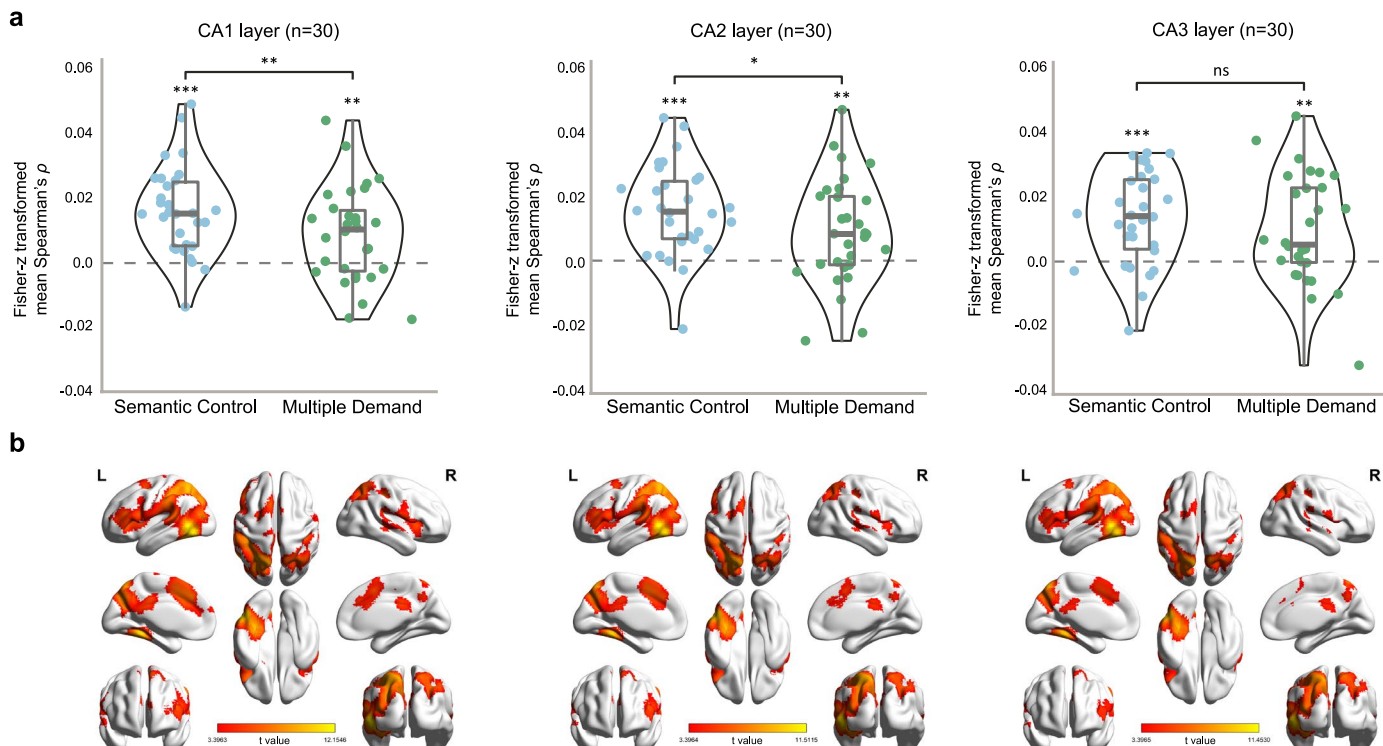

**Extended Data Fig. 3 | RSA model-group analysis results for the three layers of CA module across 30 independent model instances. a**, The results of ROI analysis, with semantic control network[25] and domain-general multi-demand (domain-general control) network[33] used as ROIs. For single-group comparisons, significance was determined by a one-tailed one-sample t-test against zero. Between-group differences were assessed using paired t-test. Asterisks indicate significance levels: **, $p < 0.01$; ***, $p < 0.001$. **b**, The whole-brain searchlight RSA results, at the threshold of voxel-level $p < 0.001$, one-tailed, cluster-level family-wise error (FWE)-corrected $p < 0.05$, highlighting the spatial patterns of model-brain correspondence across the whole brain. Each of the three subplots corresponds to one layer (CA1, CA2, and CA3) of the models.

**a**

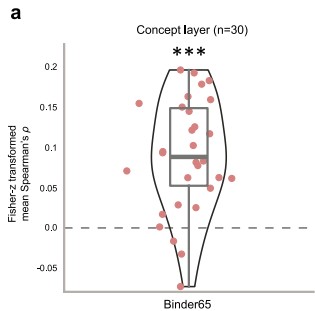

**b**

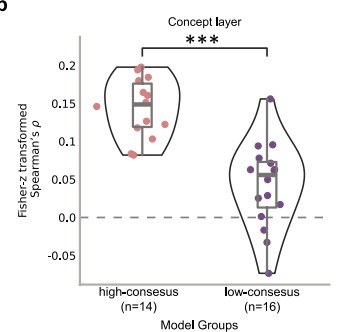

**Extended Data Fig. 4 | CATS Net-Binder65 correspondence using RSA.**
Box plots showing the Spearman's correlation coefficients (Fisher-z transformed) between our CATS Nets and the Binder65 model based on RSA analysis, controlling for the sensory input layer. **a**, The group-level results for all 30 models, with each point representing an independently trained model instance. **b**, The results of a cluster analysis dividing the models into

high-consensus (n=14) and low-consensus groups (n=16), with subsequent group-level analysis and between-group comparison. Each point in this panel also represents a model instance within the respective group. Statistical comparison between groups were performed using a two-tailed two-sample $t$-test (***, $p < 0.001$).

**a** Intermodel correlation matrix

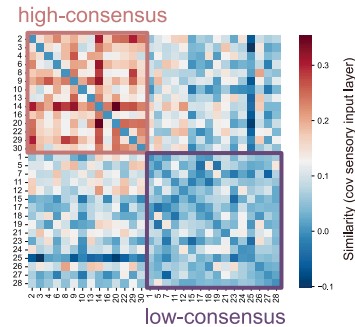

**b** ROI-level RSA with VOTC activity

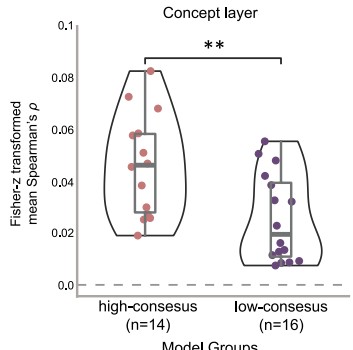

**Extended Data Fig. 5 | Clustering analysis and group-level RSA of 30 independently trained models.** Based on a two-class clustering approach, models were categorized into a high-consensus group (14/30) and a low-consensus group (16/30). Representational similarity analysis (RSA) was conducted to evaluate correspondence between each group's concept layers and human brain activity in the ventral occipitotemporal cortex (VOTC). **a**, Inter-model correlations of 30 independent trained models. Models are categorized into high-consensus (n=14) and low-consensus (n=16) groups based on representational similarity. **b**, RSA results for CATS Net's concept layer with VOTC activity. Each dot represents a single model's mean Spearman's correlation across 26 subjects. Results are plotted separately for high-consensus and low-consensus groups. Statistical comparison between groups were performed using a two-tailed two-sample t-test (**, $p < 0.01$).

# Reporting Summary

## Statistics

For all statistical analyses, confirm that the following items are present in the figure legend, table legend, main text, or Methods section.

| n/a | Confirmed | |
|---|---|---|
| ☐ | ☒ | The exact sample size ($n$) for each experimental group/condition, given as a discrete number and unit of measurement |
| ☐ | ☒ | A statement on whether measurements were taken from distinct samples or whether the same sample was measured repeatedly |
| ☐ | ☒ | The statistical test(s) used AND whether they are one- or two-sided *Only common tests should be described solely by name; describe more complex techniques in the Methods section.* |
| ☐ | ☒ | A description of all covariates tested |
| ☐ | ☒ | A description of any assumptions or corrections, such as tests of normality and adjustment for multiple comparisons |
| ☐ | ☒ | A full description of the statistical parameters including central tendency (e.g. means) or other basic estimates (e.g. regression coefficient) AND variation (e.g. standard deviation) or associated estimates of uncertainty (e.g. confidence intervals) |
| ☐ | ☒ | For null hypothesis testing, the test statistic (e.g. $F$, $t$, $r$) with confidence intervals, effect sizes, degrees of freedom and $P$ value noted *Give P values as exact values whenever suitable.* |
| ☒ | ☐ | For Bayesian analysis, information on the choice of priors and Markov chain Monte Carlo settings |
| ☒ | ☐ | For hierarchical and complex designs, identification of the appropriate level for tests and full reporting of outcomes |
| ☐ | ☒ | Estimates of effect sizes (e.g. Cohen's $d$, Pearson's $r$), indicating how they were calculated |

*Our web collection on statistics for biologists contains articles on many of the points above.*

## Software and code

Policy information about availability of computer code

| | |
|---|---|
| Data collection | No software was used |
| Data analysis | Brain Analysis: The functional images were preprocessed and analyzed using Statistical Parametric Mapping (SPM12; http://www.fil.ion.ucl.ac.uk/spm). After preprocessing, data were analysed using SPM12 and Python (version 3.10). The analysis and deep learning model codes based on Python (PyTorch 2.01) and are available on GitHub and Zenodo (https://doi.org/10.5281/zenodo.18136642). |

For manuscripts utilizing custom algorithms or software that are central to the research but not yet described in published literature, software must be made available to editors and reviewers. We strongly encourage code deposition in a community repository (e.g. GitHub). See the Nature Portfolio guidelines for submitting code & software for further information.

## Data

Policy information about availability of data

All manuscripts must include a data availability statement. This statement should provide the following information, where applicable:

- Accession codes, unique identifiers, or web links for publicly available datasets
- A description of any restrictions on data availability
- For clinical datasets or third party data, please ensure that the statement adheres to our policy

Source data are provided with this paper. The fMRI data that support the findings of this study have been deposited in the Open Science Framework (OSF) at https://osf.io/5y8p6/overview. The embeddings of SPOSE49 model are available via OSF at https://osf.io/f5rn6/files/8yjh5. Additionally, the anchor word embeddings used for the Binder65 model can be accessed at https://www.neuro.mcw.edu/index.php/resources/brain-based-semantic-representations/. We only used the ImageNet-1k training part and validation part for CATS Net training and testing in this work. Website: https://image-net.org/index.php. Website for CIFAR 100 dataset : https://www.cs.toronto.edu/~kriz/cifar.html

## Research involving human participants, their data, or biological material

Policy information about studies with human participants or human data. See also policy information about sex, gender (identity/presentation), and sexual orientation and race, ethnicity and racism.

| | |
|---|---|
| Reporting on sex and gender | Our findings apply to both sexes and genders. Sex and gender were not considered in our study design. We performed no sex- or gender-based analyses, because there was no sufficient evidence indicating differences in neural correlates of concept formation process between sexes or genders. |
| Reporting on race, ethnicity, or other socially relevant groupings | Our findings do not involve any racial or ethnic classfication. |
| Population characteristics | Participants were all right-handed and native Chinese speakers. None of them had experienced psychiatric or neurological disorders or had sustained a head injury. Twenty-nine participants (19 females; median age, 20 years; range, 18-32 years) were recruited in our study. |
| Recruitment | All participants were recruited online from colleage students in Beijing. Participant should be right-handed and nativeChinese speaker. None of them had experienced psychiatric or neurological disorders or had sustained a head injury. Eachparticipant read and signed the informed consent form before taking part in the experiments. Due to the college studentparticipants, the research results may not generalize to other populations (e.g., children). |
| Ethics oversight | All protocols and procedures of the current study were approved by the Ethics Committee of the State Key Laboratory of Cognitive Neuroscience and Learning at Beijing Normal University (ICBIR_A_0040_008). |

Note that full information on the approval of the study protocol must also be provided in the manuscript.

# Field-specific reporting

Please select the one below that is the best fit for your research. If you are not sure, read the appropriate sections before making your selection.

☐ Life sciences   ☒ Behavioural & social sciences   ☐ Ecological, evolutionary & environmental sciences

For a reference copy of the document with all sections, see nature.com/documents/nr-reporting-summary-flat.pdf

# Behavioural & social sciences study design

All studies must disclose on these points even when the disclosure is negative.

| | |
|---|---|
| Study description | This is a quantitative basic research involving human subjects. |
| Research sample | The sample sizes of the datasets was 29 (19 females; median age: 20 years; range: 18–32 years). The participants in this research are all adults, so they may not fully represent other groups (e.g., children). |
| Sampling strategy | None of the participants should have experienced psychiatric or neurological disorders or had sustained a head injury. All the participants should be all native Chinese Mandarin adult users in Beijing.<br>The sampling procedure was random designed. Sample sizes were determined by the previous model-fMRI alignment publications, and sample availability. |
| Data collection | In the fMRI experiments, participants' responses were recorded with a computer, while the ongoing brain activity during the task was recorded using a MRI scanner. The researcher was aware of the experimental conditions and the study hypothesis during data collection. |

| Timing | The fMRI dataset was collected in 2019-2020. |
|---|---|
| Data exclusions | The data of three participants were excluded from the analyses because of excessive head motion (> 3 mm/3°). |
| Non-participation | No participants declined participation or dropped out. |
| Randomization | Participants were not allocated into experimental groups. |

# Reporting for specific materials, systems and methods

We require information from authors about some types of materials, experimental systems and methods used in many studies. Here, indicate whether each material, system or method listed is relevant to your study. If you are not sure if a list item applies to your research, read the appropriate section before selecting a response.

## Materials & experimental systems

| n/a | Involved in the study |
|---|---|
| ☒ | ☐ Antibodies |
| ☒ | ☐ Eukaryotic cell lines |
| ☒ | ☐ Palaeontology and archaeology |
| ☒ | ☐ Animals and other organisms |
| ☒ | ☐ Clinical data |
| ☒ | ☐ Dual use research of concern |
| ☒ | ☐ Plants |

## Methods

| n/a | Involved in the study |
|---|---|
| ☒ | ☐ ChIP-seq |
| ☒ | ☐ Flow cytometry |
| ☐ | ☒ MRI-based neuroimaging |

## Plants

| Seed stocks | *Report on the source of all seed stocks or other plant material used. If applicable, state the seed stock centre and catalogue number. If plant specimens were collected from the field, describe the collection location, date and sampling procedures.* |
|---|---|
| Novel plant genotypes | *Describe the methods by which all novel plant genotypes were produced. This includes those generated by transgenic approaches, gene editing, chemical/radiation-based mutagenesis and hybridization. For transgenic lines, describe the transformation method, the number of independent lines analyzed and the generation upon which experiments were performed. For gene-edited lines, describe the editor used, the endogenous sequence targeted for editing, the targeting guide RNA sequence (if applicable) and how the editor was applied.* |
| Authentication | *Describe any authentication procedures for each seed stock used or novel genotype generated. Describe any experiments used to assess the effect of a mutation and, where applicable, how potential secondary effects (e.g. second site T-DNA insertions, mosiacism, off-target gene editing) were examined.* |

## Magnetic resonance imaging

### Experimental design

| Design type | Task-based fMRI (block design). |
|---|---|
| Design specifications | Ninety-five objects were chosen, including 3 common domains (32 animals, 35 small manipulable artefacts, and 28 large nonmanipulable artefacts). Each object was presented as a 400 × 400 pixels coloured image displaying a representative exemplar against a white background (10.55° × 10.55° of visual angle). All the participants were asked to name each displayed picture using oral language. The whole experiment included 6 runs, with each item repeated for 6 times across the experiment. Each run (8 min 45 s) consisted of 95 trials, with each item presented once per run. The trial structure consisted of a 0.5 s fixation, followed by a 0.8 s stimulus presentation and an intertrial interval (ITI) ranging from 2.7 s to 14.7 s. |
| Behavioral performance measures | There is no behaviour performance measures in this study. |

### Acquisition

| Imaging type(s) | Functional, structural |
|---|---|
| Field strength | 3 Tesla |
| Sequence & imaging parameters | Functional and anatomical MRI images were collected at the MRI center, Beijing Normal University using a $3$ Tesla Siemens Trio Tim Scanner. A high-resolution 3D structural image was collected with a 3D magnetisation prepared-rapid gradient echo (3D-MPRAGE) sequence in the sagittal plane (144 slices, TR = 2530 ms, TE = 3.39 ms, flip angle = 7°, matrix size = 256 × 256, voxel size = 1.33 × 1 × 1.33 mm). Functional images were acquired with an echo-planar imaging |

(EPI) sequence (33 axial slices, TR = 2000 ms, TE = 30 ms, flip angle = 90°, matrix size = 64 × 64, voxel size = 3 × 3 × 3.5 mm with a gap of 0.7 mm).

Area of acquisition | A whole brain scan.

Diffusion MRI | ☐ Used  ☒ Not used

## Preprocessing

Preprocessing software | The functional images were preprocessed and analyzed using Statistical Parametric Mapping (SPM12; http://www.fil.ion.ucl.ac.uk/spm).

Normalization | The images were normalized to Montreal Neurological Institute (MNI) space via unified segmentation (resampling into 3 × 3 × 3 mm^3 voxel size).

Normalization template | MNI305

Noise and artifact removal | For the preprocessing of the task fMRI data, the first five volumes of each functional run were discarded to reach signal equilibrium. Slice timing and 3-D head motion correction were performed. After normalization, the functional images were spatially smoothed using a 6-mm full-width-half-maximum Gaussian kernel for univariate analysis but not for multivariate pattern analysis. Temporal bandpass filtering (0.01–0.1 Hz) was performed to reduce the effects of high-frequency noises.

Volume censoring | None

## Statistical modeling & inference

Model type and settings | Multivariate pattern analysis (MVPA); GLM analysis was first performed to obtain results of each regressors; during GLM analysis, six head motion parameters were included as nuisance regressors, and a high-pass filter (128 s) was used to remove low-frequency signal drift for each run; then MVPA were performed; the results of MVPA were entered into second-level (between-subject) random-effect analysis.

Effect(s) tested | Representational similarity analyses; partial Spearman's correlation.

Specify type of analysis: | ☐ Whole brain  ☐ ROI-based  ☒ Both

Anatomical location(s) | The VOTC mask was defined as regions showing stronger activation to all pictures relative to the rest in the fMRI dataset (FDR q < 0.05) within the cerebral mask combining the posterior and temporooccipital divisions of inferior temporal gyrus (15#, 16#), the inferior division of lateral occipital cortex (23#), the posterior division of parahippocampal gyrus (35#), the lingual gyrus (36#), the posterior division of temporal fusiform cortex (38#), the temporal occipital fusiform cortex (39#), the occipital fusiform gyrus (40#), the supracalcarine cortex (47#) and the occipital pole (48#) in the Harvard-Oxford Atlas (probability > 0.2).

Statistic type for inference

(See Eklund et al. 2016) | All effects in whole-brain level were tested by one sample t-tests and cluster-wise FWE correction as implemented in SPM12. Effects at ROI level were tested by null-hypothesis one, pair and independent sample ttests.

Correction | For whole-brain analysis, multiple comparison corrections were conducted using cluster-level FWE correction (p <.05) asimplemented in SPM12 (voxel-wise p <.001).

## Models & analysis

| n/a | Involved in the study |
| --- | --- |
| ☒ ☐ | Functional and/or effective connectivity |
| ☒ ☐ | Graph analysis |
| ☐ ☒ | Multivariate modeling or predictive analysis |

Multivariate modeling and predictive analysis | We conducted searchlight MVPA within a ventral occipitotemporal cortex (VOTC) mask. This mask was defined on the basis of regions showing stronger activation to all pictures relative to baseline in hearing participants from the fmri dataset (q < 0.05, FDR-corrected). The functional mask was further constrained by the following anatomical parcels (Harvard–Oxford Atlas, probability > 0.2): the posterior and temporooccipital divisions of the inferior temporal gyrus (15#, 16#), the inferior division of the lateral occipital cortex (23#), the posterior division of the parahippocampal gyrus (35#), the lingual gyrus (36#), the posterior division of the temporal fusiform cortex (38#), the temporal occipital fusiform cortex (39#), the occipital fusiform gyrus (40#), the supracalcarine cortex (47#), and the occipital pole (48#). This definition resulted in the selection of 2467 voxels in the left hemisphere and 2420 voxels in the right hemisphere.

For each voxel within the VOTC mask, multivariate activation patterns within a sphere (radius = 10 mm) centred at that voxel were extracted. Neural RDMs were computed with the Pearson distance within the searchlight sphere. Then, Spearman's rank correlation coefficients between the neural RDM and model-derived concept RDMs was computed, controlling for the effects of sensory input layer. Correlation maps were obtained for each participant by moving the searchlight centre across the VOTC mask. These maps

were Fisher z-transformed and spatially smoothed with a 6 mm full-width half-maximum (FWHM) Gaussian kernel. The correlation maps were compared to 0 with one-tailed one-sample t tests.

ROI-based MVPA was conducted with the same VOTC mask, semantic control mask and multiple demand network mask as ROIs. Specifically, multivariate activity patterns for each stimulus within the ROI mask were extracted. Neural RDMs were computed on the basis of Pearson distances and then correlated with the RDM generated by conceptl layer and CA layers. The resulting correlation coefficients between the neural and model RDMs were Fisher z-transformed and compared to zero with one-tailed one-sample t tests.

