## [Peer Review file · Nature Computational Science]

A neural network for modeling human concept formation, understanding and communication

Corresponding Author: Professor Shan Yu

Version 0:

Decision Letter:

** Please ensure you delete the link to your author homepage in this e-mail if you wish to forward it to your co-authors. **

Dear Professor Yu,

Your manuscript "A neural network model for concept formation, understanding and communication" has now been seen by 2 referees, whose comments are appended below. You will see that while they find your work of interest, they have raised points that need to be addressed before we can make a decision on publication.

The referees' reports seem to be quite clear. Naturally, we will need you to address **all** of the points raised.

While we ask you to address all of the points raised, the following points need to be substantially worked on:

- Please include ablation studies and hyperparameter explorations.
- Numerous experiments are presented, but the demonstrations are often minimal. Stronger baselines should be included.
- Please include a clear comparison and explanation, and the rationale behind generating a new network by combining existing ones.
- Discuss if this new model can generalize to domains outside visual classification.
- The manuscript should explain whether the translation module maintains semantic detail.
- Please consider real human-generated data, and discuss how the representations differ between concrete and nonconcrete concepts.

Please use the following link to submit your revised manuscript and a point-by-point response to the referees' comments (which should be in a separate document to any cover letter):

Link Redacted

** This url links to your confidential homepage and associated information about manuscripts you may have submitted or be reviewing for us. If you wish to forward this e-mail to co-authors, please delete this link to your homepage first. **

To aid in the review process, we would appreciate it if you could also provide a copy of your manuscript files that indicates your revisions by making use of Track Changes or similar mark-up tools. Please also ensure that all correspondence is marked with your Nature Computational Science reference number in the subject line.

In addition, please make sure to upload a Word Document or LaTeX version of your text, to assist us in the editorial stage.

To improve transparency in authorship, we request that all authors identified as 'corresponding author' on published papers create and link their Open Researcher and Contributor Identifier (ORCID) with their account on the Manuscript Tracking System (MTS), prior to acceptance. ORCID helps the scientific community achieve unambiguous attribution of all scholarly contributions. You can create and link your ORCID from the home page of the MTS by clicking on 'Modify my Springer Nature account'. For more information please visit please visit <a

<http://www.springernature.com/orcid>>www.springernature.com/orcid.

We hope to receive your revised paper within three weeks. If you cannot send it within this time, please let us know.

Best regards,

Ananya Rastogi, PhD
Senior Editor
Nature Computational Science

Reviewers comments:

Reviewer #1 (Remarks to the Author):

This paper introduces a novel neural network strategy for visual concept formation. A low-dimensional concept space is trained to represent visual classes from a standard dataset (ImageNet or CIFAR), in such a way that it can condition a decision network to solve a present/absent binary task for each specific class. The authors argue that this system bears similarity with human concept learning. The learned concept space is more functionally specialized than a random space. It is possible to transfer a novel class concept from one instance of the concept space (the teacher) to another (the student). Finally, the concept space is correlated with representations in human occipito-temporal (visual) cortex, while the concept-abstraction layers (modulating the decision network) appear correlated with the human semantic control network.

There are interesting ideas in this work, and some potentially promising results. On the other hand, many of the claims have only weak support because of a lack of comparison baselines and systematic ablation tests (see below). It is also unclear what the main message should be for general audiences: it is not a revolutionary machine learning method that surpasses state-of-the-art on any benchmark; it is also not a fully-fleshed model of human concept learning. My feeling is that the paper's interest lies somewhere in-between, as a vaguely promising avenue of research to model and understand (in a remote future) human concept learning.

Below is a list of issues that could be addressed to make the results and conclusions more convincing—without necessarily making the paper more broadly appealing; for this, I unfortunately have no suggestion...

1. A lot of the text gives the impression that the concept formation in CATS Net is autonomous (lines 115, 639) or self-organized (line 53). To me, this suggests a form of self-supervised or unsupervised learning, whereas in practice the network training is entirely supervised by the human-made class labels (each dataset image is associated with a given class label and a yes/no decision). I found this misleading.
2. The network architecture is presented without any empirical justification for the choices made. Ablations and/or hyperparameter explorations would help alleviating the impression of arbitrariness. Examples of choices to be substantiated include the ResNet feature extractor (is this crucial for the results, or would other visual backbones work just as well?), the dimensionality of the concept space (why 20 dimensions? Would it still work with 10 or with 100 dimensions?), the number of CA/TS layers (why 3?), and the use of an alternating two-phase training (Why not use end-to-end training of the concept space and CA/TS layers?).
3. Numerous experiments are presented, but the demonstrations are often minimal, i.e. comparisons against worst-case scenarios (e.g. accuracy above chance, correlations above zero). Overall, this conveys the impression that the proposed strategy is better than applying random concept vectors, a very low bar to pass. Stronger baselines should be considered whenever possible. For example, the concept space could be replaced by a fixed one-hot class vector, or a fixed Word2Vec vector representation. Or the concept space could be randomly initialized as currently, and then kept frozen during CA/TS layer training. Maybe the CA/TS layers themselves would re-organize in such a way as to exploit the chosen concept dimensions, and similar or better accuracy would be observed? The same can be said of the “teacher-student transfer” experiment, where CATS Net accuracy is above chance level: how would it compare to either of the baselines described above? We already see by comparing Figures 3e and 4c that the Word2vec concept space provides better transfer than CATS Net. The authors interpret this result as a positive sign that CATS Net is learning meaningful vectors, but this could just as easily be taken as a negative sign that CATS Net provides no advantage against existing concept extraction methods.
4. I was very confused upon reading Method section 4.8 “Leave-one-out experiment and new concept abstraction”, because the description did not seem to correspond to any experiment or figure. Then I realized (in Method section 4.9) that some methodological tricks for data augmentation were used in some of the experiments (translation module) without any explanation in the main text.
5. What was the dimensionality of the pretrained Word2vec representation? Was it 20, or did you project it down to 20, and in this case, from what original dimension (300?), and what was the projection method (PCA)?

Reviewer #2 (Remarks to the Author):

The manuscript titled “A neural network model for concept formation, understanding, and communication” by Guo et al. introduces CATS Net, a dual-module neural network architecture that combines concept abstraction and sensorimotor task-solving within a single computational framework. The authors show that this model not only creates low-dimensional concept representations but also restores sensorimotor functions through hierarchical gating, which aids in concept transfer and cross-model communication. They also discuss how the emergent representations of the model align with human neurosemantic spaces and brain response patterns in VOTC and semantic control regions. While this work offers an interesting approach by combining two types of established networks to explore new research directions, it is not clear what specific advancements this model provides over previous networks in terms of its goals and applications. Additionally, the assumptions of the model and other methodological details need further clarification.

Major Points:

1. It seems the new model is designed for image classification with concept constraints. I had trouble fully appreciating the novelty of this creation. What is the difference between this creation and the classic ImageNet-based convolutional neural networks with attention modules? Will this network outperform existing networks in terms of image classification accuracy or computational resources? How is this network more advantageous than others? How is this network different and more advanced than the communication protocols used in multi-agent reinforcement learning? Without a clear comparison and explanation, I find it difficult to see the benefits and the rationale behind generating a new network by combining existing ones.
2. This network is entirely vision-based, adding an extra question mark to the difference from well-trained and well-developed vision models. Is it possible for this new model to generalize to domains outside visual classification (e.g., auditory or embodied tasks), as these are also important for conceptual learning? Can this model be applied to temporally dynamic contexts?
3. The idea that concept vectors can be communicated between independently trained networks is interesting. However, the manuscript could explain whether the translation module maintains semantic detail. Is any information lost during translation, especially when concepts involve subtle or overlapping features?
4. The authors used Word2Vec to summarize the human concept space. While Word2Vec is a powerful tool for exploring the human concept space, the concept range in this study is limited and focused only on concrete concepts. It would be better if real human-generated data could be considered, and the authors should discuss how the representations differ between concrete and nonconcrete concepts.
5. The matrices in Figure 4a look very different from each other, and the correlations are low. Is the significance mainly caused by the large number of features?
6. The model-brain correlations are low too (< 0.1). What do the noise ceilings look like across the brain? Do the correlations mainly reflect differences in noise ceilings? How robust are these correlations across varying model initializations, dataset splits, or semantic ontologies? Were multiple neurosemantic models tested, and how does Binder65 compare to alternatives like WordNet-based semantic spaces?
7. While CAM-based visualizations are shown, providing more details on the interpretability of the 20-dimensional concept space would be helpful. Are there prominent dimensions that can be labeled semantically (e.g., animacy, shape, size)?
8. The authors note that the CA module aligns with the semantic control network. Is there any previous research connecting multiplicative gating (as used here) to control regions such as IFG or dmPFC?

Minor Points:

1. The paper makes strong claims about “human-like” communication. While partly justified, the authors might consider softening this claim or clarifying that “symbolic transfer” does not yet equal natural language communication.

Reviewer #2 (Remarks on code availability):

I reviewed the code provided by the authors. The repository is well-documented and clearly organized, with modular and readable code that closely aligns with the methodological descriptions in the paper.

The repository includes a comprehensive README file, which outlines the installation steps, dependencies, and usage instructions in a straightforward manner. From inspecting the code and the structure of the provided scripts and configuration files, it is clear that the authors have made a serious effort to ensure usability and reproducibility. Although I did not execute the code, the structure and documentation suggest that the results should be reproducible with the provided instructions and standard computational resources.

Overall, I think the codebase is a valuable resource for the community and should allow researchers to validate, reuse, and build upon the authors' proposed CATS Net architecture.

Version 1:

Decision Letter:

Our ref: NATCOMPUTSCI-25-1637A

14th November 2025

Dear Dr. Yu,

Thank you for submitting your revised manuscript "A neural network for modeling human concept formation, understanding and communication" (NATCOMPUTSCI-25-1637A). It has now been seen by the original referees and their comments are below. The reviewers find that the paper has improved in revision, and therefore we'll be happy in principle to publish it in Nature Computational Science, pending minor revisions to satisfy the referees' final requests and to comply with our editorial and formatting guidelines.

We are now performing detailed checks on your paper and will send you a checklist detailing our editorial and formatting requirements in about 2 weeks. Please do not upload the final materials and make any revisions until you receive this additional information from us.

TRANSPARENT PEER REVIEW

Nature Computational Science offers a transparent peer review option for original research manuscripts. We encourage increased transparency in peer review by publishing the reviewer comments, author rebuttal letters and editorial decision letters if the authors agree. Such peer review material is made available as a supplementary peer review file. **Please remember to choose, using the manuscript system, whether or not you want to participate in transparent peer review.**

Thank you again for your interest in Nature Computational Science. Please do not hesitate to contact me if you have any questions.

Sincerely,
Fernando

--

Fernando Chirigati, PhD
Chief Editor, Nature Computational Science
Nature Portfolio

ORCID

Author names using non-Roman characters

Nature Portfolio journals can support presentation of author names using non-Roman characters in the HTML version of the article. If you wish to, please include author names in parentheses after the Roman-character spelling; [see example online here](https://www.nature.com/articles/s44222-024-00258-2). Currently supported scripts are: Arabic, Chinese, Cyrillic, Devanagari, Greek, Hebrew, Hangul, Japanese and Persian. You will be asked to verify the rendering is correct at proof stage.

Reviewer #1 (Remarks to the Author):

I have read the revised manuscript as well as the rebuttal letter. I commend the authors for undergoing significant revisions that improved a number of aspects. In particular, the architectural choices and hyperparameters no longer appear arbitrary, but are now justified with systematic explorations and ablations. The motivations for the study have also been clarified, and remain commensurate with the results obtained. As I explained in the previous round, the paper is neither a major advance in AI, nor a complete model of human concept formation, but rather lies somewhere in between. I nonetheless believe that the potential contribution to both fields (AI and human neuro-psychology) is meaningful, and that the paper's merits outweigh its shortcomings.

Reviewer #1 (Remarks on code availability):

Same as previous round.

Reviewer #2 (Remarks to the Author):

The authors have successfully addressed all my questions.

Version 2:

Decision Letter:

Dear Professor Yu,

We are pleased to inform you that your Article "A neural network for modeling human concept formation, understanding and communication" has now been accepted for publication in Nature Computational Science.

Once your manuscript is typeset, you will receive an email with a link to choose the appropriate publishing options for your paper and our Author Services team will be in touch regarding any additional information that may be required.

Authors may need to take specific actions to achieve compliance with funder and institutional open access

mandates. If your research is supported by a funder that requires immediate open access (e.g. according to [Plan S principles](https://www.springernature.com/gp/open-science/plan-s-compliance) or the [NIH public access policy](https://www.springernature.com/gp/open-science/us-federal-agency-compliance)) then you should select the gold OA route, and we will direct you to the compliant route where possible. Because authors warrant under our subscription licensing terms that they haven't committed to licensing any version of their article under a licence inconsistent with the terms of our agreement – including the applicable embargo period – publication under the subscription model isn't suitable for authors whose funders require no embargo.

Acceptance of your manuscript is conditional on all authors' agreement with our publication policies (see <https://www.nature.com/natcomputsci/for-authors>). In particular your manuscript must not be published elsewhere and there must be no announcement of the work to any media outlet until the publication date (the day on which it is uploaded onto our web site).

Before your manuscript is typeset, we will edit the text to ensure it is intelligible to our wide readership and conforms to house style. We look particularly carefully at the titles of all papers to ensure that they are relatively brief and understandable.

Once your manuscript is typeset, you will receive a link to your electronic proof via email with a request to make any corrections within 48 hours. If, when you receive your proof, you cannot meet this deadline, please inform us at rjsproduction@springernature.com immediately.

If you have queries at any point during the production process then please contact the production team at rjsproduction@springernature.com.

We welcome the submission of potential cover material (including a short caption of around 40 words) related to your manuscript; suggestions should be sent to Nature Computational Science as electronic files (the image should be 300 dpi at 210 x 297 mm in either TIFF or JPEG format). We also welcome suggestions for the Hero Image, which appears at the top of our [home page](http://www.nature.com/natcomputsci); these should be 72 dpi at 1400 x 400 pixels in JPEG format. Please note that such pictures should be selected more for their aesthetic appeal than for their scientific content, and that colour images work better than black and white or grayscale images. Please do not try to design a cover with the Nature Computational Science logo etc., and please do not submit composites of images related to your work. I am sure you will understand that we cannot make any promise as to whether any of your suggestions might be selected for the cover of the journal.

Best regards,
Fernando (on behalf of Ananya Rastogi)

--

Fernando Chirigati, PhD
Chief Editor, Nature Computational Science
Nature Portfolio

P.S. Click on the following link if you would like to recommend Nature Computational Science to your librarian: https://www.springernature.com/gp/librarians/recommend-to-your-library

** Visit the Springer Nature Editorial and Publishing website at www.springernature.com/editorial-and-publishing-jobs for more information about our career opportunities. If you have any questions please click here.**

Dear Reviewers,

We thank all reviewers for your insightful comments. In the following, we begin with our summarized *general response* to all reviewers, and then we provide a point-by-point reply to both reviewers' original comments.

Reviewer #1: page 2 - 11

Reviewer #2: page 12 - 29

Appendix: page 31 - 33

We have provided you with two versions of the manuscript for review. One is the highlight revised version, where we have marked the modifications corresponding to each issue in blue font. **The line numbers and section references in the following replies are based on this version.** Additionally, we have also provided a Track Changes version generated by the *latexdiff* command to compare the changes between our revised version and the initial one.

Given the extensive supplementary results, and to ensure the clarity of all figures which can be compromised by MS Word's image compression, we have provided all relevant files in the attached ZIP archive for your convenience.

General Response:

Clarification of our main message:

We thank both reviewers for acknowledging the interest of our work. The key concern was that the main message was not sufficiently clear; accordingly, we have added analyses and clarifications to make the model's goals and applications explicit. Therefore, we believe it is necessary to clarify before replying the point-by-point comments.

First, CATS Net is not specifically designed as an advanced image classification system, but rather as a computational model for cognitive neuroscience to understand the mechanistic foundation of human concept learning. In the manuscript, we have demonstrated why the gating mechanism underlying CATS Net might be the computational mechanism of cognitive control, utilizing brain imaging data (*Results* Section 2.5: VOTC and semantic control networks) and evidence from cognitive psychology model (*Results* Section 2.2 & 2.6: Binder65). Furthermore, based on all reviewers' feedback, we have specifically reinforced the dimensional analysis regarding the formation of CATS Net's conceptual space and comparisons with additional real human data using THINGS and SPOSE49 model. Taken together, we believe these analyses will strengthen our main message, i.e., we build CATS Net to model and understand the computational mechanism of human concept learning.

Second, for artificial intelligence, CATS Net provide a new framework for treating a concept as an executable, transferable function (e.g., a binary discrimination skill or else) that can be compressed into a low-dimensional vector and communicated, enabling the recipient model/agent to acquire new capabilities without direct experience. In addition to the evidence in the visual domain already presented in the article, we have now also provided evidence in the non-visual domain. CATS Net generalizes to domains where concepts are not pre-specified and must be derived as compact representations of complex task-level functions (e.g., policies). We

designed a reinforcement learning task in a multi-maze scenario, where five CATS Net-based agents explore different mazes separately and form concepts of maze-solving strategies. Then, through communication, they can directly obtain strategies to exit the maze without undergoing training or direct experience (see Appendix of this letter for details). This is currently unachievable using CNNs with attention modules, ViTs, or in Multi-Agent Reinforcement Learning (MARL).

Reviewer #1 (Remarks to the Author):

This paper introduces a novel neural network strategy for visual concept formation. A low-dimensional concept space is trained to represent visual classes from a standard dataset (ImageNet or CIFAR), in such a way that it can condition a decision network to solve a present/absent binary task for each specific class. The authors argue that this system bears similarity with human concept learning. The learned concept space is more functionally specialized than a random space. It is possible to transfer a novel class concept from one instance of the concept space (the teacher) to another (the student). Finally, the concept space is correlated with representations in human occipito-temporal (visual) cortex, while the concept-abstraction layers (modulating the decision network) appear correlated with the human semantic control network.

Reply:

We sincerely appreciate your summary, which effectively captures the core contributions of our work.

There are interesting ideas in this work, and some potentially promising results. On the other hand, many of the claims have only weak support because of a lack of comparison baselines and systematic ablation tests (see below). It is also unclear what the main message should be for general audiences: it is not a revolutionary machine learning method that surpasses state-of-the-art on any benchmark; it is also not a fully-fleshed model of human concept learning. My feeling is that the paper's interest lies somewhere in-between, as a vaguely promising avenue of research to model and understand (in a remote future) human concept learning.

Reply:

We sincerely appreciate your insightful and constructive feedback, which has helped us better clarify the positioning and contributions of our work. First, we acknowledge limitations in our initial presentation regarding comparison baselines and ablation studies. As detailed in responses and the revised manuscript, we have supplemented comprehensive experiments to address these issues. Second, we apologize for failing to clearly articulate our core message. Our primary goal is not to develop a state-of-the-art image classification model. Instead, we aim to propose a computational framework that captures the core properties of semantic cognition and human concept learning, which aligns with human from both behaviour functionality and neuroimages (also see general response). Specifically, it models how low-dimensional, transferable concept representations emerge from sensorimotor experience and interact with cognitive control mechanisms. By focusing on the emergence of functional concept structures (rather than task performance alone) and validating them against human behavioral and neural data, we aim to provide a mechanistic understanding for human concept learning. This claim is strengthened by new evidence and experiments. Details are provided in the subsequent point-by-point responses and revised manuscript sections. Thank you again for guiding us to strengthen these aspects. We believe these revisions better convey the significance of our work.

Below is a list of issues that could be addressed to make the results and conclusions more convincing—without

necessarily making the paper more broadly appealing; for this, I unfortunately have no suggestion...

Reply:

We have implemented new analyses to better address this issue. Your kind assistance has helped us better clarify the novelty and significance of our work. Please see below for details.

Comment 1. A lot of the text gives the impression that the concept formation in CATS Net is autonomous (lines 115, 639) or self-organized (line 53). To me, this suggests a form of self-supervised or unsupervised learning, whereas in practice the network training is entirely supervised by the human-made class labels (each dataset image is associated with a given class label and a yes/no decision). I found this misleading.

Reply:

We thank you for this insightful comment. We agree that our use of terms like “autonomous” and “self-organized” could be misleading, and we apologize for the confusion. Our training process is indeed fully supervised, relying on human-annotated class labels.

In the original manuscript, we aimed to highlight that the structure of the concept vector space is emergent and not explicitly pre-defined or encoded by the researchers. The model learns from pixels and labels to organize a low-dimensional concept space on its own. This emergent structure is meaningful, as evidenced by its significant correlation with human semantic spaces (e.g., Word2Vec derived from linguistic data) and its ability to support knowledge transfer across different networks (i.e., *Results* Section 2.3). This suggests the model captures some intrinsic principles of conceptual organization.

According to your suggestion, we removed words such as "autonomous" and replaced words such as "self-organizes" with "develops", "learned", etc.

We revised the relevant text in our manuscript:

(*Abstract*: line 50 - 52)

“Crucially, the system develops transferable semantic structure based on concept representations that enable cross-network knowledge transfer through conceptual communication.”

(*Introduction* Section 1: line 115 - 119)

“In the CATS Net, conceptual formation is modeled as the CA module forming a low-dimensional input space of concept vector, while concept understanding is modeled by a gating mechanism using the conceptual vectors to dynamically reconfigure the TS module.”

(*Results* Section 2.1: line 210 - 213)

“For all unseen images from 1000 categories tested on ImageNet-1k dataset, the learned concept vectors for each category achieved a judgement accuracy ranging from 0.86 to 1.00, well above the chance level of 0.5 (Fig 2a).”

(*Results* Section 2.1: line 248 - 251)

“These results indicate that a highly compressed concept vector (20-dim real vector) can be generated and

used to configure a complex network to perform specific tasks, suggesting a promising framework for artificial neural networks to emerge functionally meaningful low-dimensional concept space."

(Discussion Section 3: line 813 - 815)

"The first key advance is CATS Net's ability to develop functionally meaningful concept representations, unlike systems reliant on human-curated symbols (e.g., MLLMs)."

Comment 2. The network architecture is presented without any empirical justification for the choices made. Ablations and/or hyperparameter explorations would help alleviating the impression of arbitrariness. Examples of choices to be substantiated include the ResNet feature extractor (is this crucial for the results, or would other visual backbones work just as well?), the dimensionality of the concept space (why 20 dimensions? Would it still work with 10 or with 100 dimensions?), the number of CA/TS layers (why 3?), and the use of an alternating two-phase training (Why not use end-to-end training of the concept space and CA/TS layers?).

Reply:

We thank you for this important question about the empirical justification of our architectural choices. We have conducted comprehensive ablation studies to validate the robustness and generalizability of our CATS Net architecture across different hyperparameter settings. The result figure is shown below:

1. Backbone architecture:

We tested our framework with different visual backbones beyond ResNet50, including Vision Transformer (ViT-B/16). The results demonstrate that CATS Net maintains consistent performance across different feature extractors, achieving similar accuracy (0.966, the third bar from the left in Reply Fig.1) regardless of the backbone choice. This indicates that our hierarchical gating mechanism is robust and not dependent on specific CNN architectures.

2. Concept space dimensionality:

First, our selection of a 20-dim concept space is grounded in established neuroscientific literature on human conceptual representations. As we described in our manuscript, "*First is concept formation: The higher-dimensional sensory-motor experience is compressed into lower-dimensional representational spaces [2–5], whose dimensionality typically ranges from 20 to several hundreds [6–10].*" (**Introduction Section 1: line 80-83**). This neurobiological evidence suggests that human concept formation involves compressing higher-dimensional sensory-motor experience into lower-dimensional representations, with 20 dimensions falling well within the empirically observed range for effective conceptual encoding in biological systems. Our choice thus aligns with the natural dimensionality constraints observed in human conceptual processing.

Second, we empirically validated this architectural choice through systematic ablation studies. We conducted experiments across different concept space dimensions (10, 20, and 100 dimensions) and found that performance remained remarkably consistent, with 20-dimension providing an optimal balance between representational efficiency and compression capability (the second, fourth and fifth bar from the left in Reply Fig.1). The convergence between neurobiological evidence and our empirical findings strengthens the theoretical foundation of our approach.

3. Network depth:

We tested CA/TS modules with 1, 3, and 5 layers. The results demonstrate that our framework is remarkably robust to depth variations, with all configurations achieving comparable performance (the sixth and seventh bar from the left in Reply Fig.1).

4. Training strategy:

We compared our alternating two-phase training with end-to-end joint training. Interestingly, both approaches yielded nearly identical results, suggesting that the concept formation process is robust to training methodology (the last bar from the left in Reply Fig.1). We still want to retain the two-stage training approach, which facilitates readers' understanding. The learning of concept space can be independent from the learning of network parameters. We believe this approach aids readers in understanding that once CA/TS has been trained, new functional and meaningful network configurations can be directly obtained by acquiring concept vectors solely in the concept space. As demonstrated in our leave-one-out experiments and communication experiments, this process does not involve modifications to network parameters, but only involves operations in the concept space.

Reply Fig. 1. Ablation studies and hyperparameter explorations. The left most 2 bars was adopted from Fig. 2a, while the others represent the average of mean accuracy across 5 independently initialized models after training, and each point represents the corresponding mean accuracy cross all categories. The Random and learned bars represents the average of mean accuracy across 30 models before and after training (with the selection of a pretrained ResNet50 as the feature extractor, a concept size of 20, 3-layer CA/TS modules, and two-phase training), and each pair point represents the corresponding mean accuracy of each category.

These comprehensive ablation studies demonstrate that CATS Net's robustness, which indicates that our framework captures a general computational principle for concept formation that is broadly applicable across different implementation details. Reply Fig. 1 has been included as Figure S1a in the Supplementary Information. The corresponding results have also been integrated into the article, as shown below:

We revised the relevant text in our manuscript:

(Results Section 2.1: line 190 - 192)

“The framework's design is agnostic to the backbone architecture, demonstrating robust generalization across

different structures like ResNet50 [16] and ViT-B/16 [17] (see Fig S1a).”

(*Results* Section 2.1: line 204 - 210)

“This two-phase training strategy, along with the selection of a pretrained ResNet50 as the feature extractor, a concept size of 20, and 3-layer CA/TS modules, was validated as optimal through ablation studies (see Fig S1a and Supplementary Information). Using this established configuration, we trained 30 independently initialized models on the ImageNet-1k dataset, which successfully generated a set of visual concept vectors for task solving.”

(*Methods* Section 4.2: line 913 - 930)

“In concept abstraction task on image dataset, we use a 20-dim real-valued vector to present each category. This dimension was selected from a tested range of {10, 20, 100}, as it offered the optimal trade-off between compression efficiency and representational capacity. The compactness of this vector space, compared to the high-dimensional parameter space of the neural network, reflects the highly compressed nature of the concepts. The model's pipeline begins with a pretrained ResNet50 backbone, chosen over ViT for its computational efficiency after observing similar performance from both (We use official V1 weights from PyTorch for both backbones). The extracted 2,048-dim features are then fed into the TS module. This module is a 3-layer perceptron ([2,048-100-100-2]) with batch normalization and ReLU activation. The 3-layer architecture was adopted for its demonstrated robustness, as our tests with 1, 3, and 5 layers all yielded comparable performance (see Fig S1a and Supplementary Information). To match this structure, the CA module is also a 3-layer perceptron ([20-2,048-100-100]), which takes the 20-dim concept vector as input and uses the Sigmoid function to generate controlling signals between 0 and 1. The output layer of the TS module consists of two neurons for "Yes" (0, 1) and "No" (1, 0) classification, optimized using a cross-entropy loss.”

(*Methods* Section 4.3: line 978 - 982)

“It provides better interpretability of concept learning dynamics, which implies the learning of concept space can be independent from the learning of network parameters. We also validated this approach against end-to-end joint training and found comparable performance (Fig S1a), confirming that the concept formation process is robust to training methodology.”

Comment 3. Numerous experiments are presented, but the demonstrations are often minimal, i.e. comparisons against worst-case scenarios (e.g. accuracy above chance, correlations above zero). Overall, this conveys the impression that the proposed strategy is better than applying random concept vectors, a very low bar to pass. Stronger baselines should be considered whenever possible. For example, the concept space could be replaced by a fixed one-hot class vector, or a fixed Word2Vec vector representation. Or the concept space could be randomly initialized as currently, and then kept frozen during CA/TS layer training. Maybe the CA/TS layers themselves would re-organize in such a way as to exploit the chosen concept dimensions, and similar or better accuracy would be observed? The same can be said of the “teacher-student transfer” experiment, where CATS Net accuracy is above chance level: how would it compare to either of the baselines described above? We already see by comparing Figures 3e and 4c that the Word2vec concept space provides better transfer than CATS Net. The authors interpret this result as a positive sign that CATS Net is learning meaningful vectors, but this could just as easily be taken as a negative sign that CATS Net provides no advantage against existing concept extraction methods.

Reply:

We sincerely thank you for this insightful suggestion. We agree that providing stronger baselines is crucial for rigorously evaluating the properties of the concept space generated by CATS Net. **On the one hand (point 1 below)**, following your advice, we have conducted a series of ablation studies to compare our model with several alternative methods for constructing the concept space. The results demonstrate the advantages of our approach while revealing important trade-offs between task performance and the semantic properties of the representation. **On the other hand (point 2 below)**, we provide a more specific explanation of how CATS model captures the core processes underlying human concept learning.

1. Ablation studies on concept space construction:

We compared the performance of different types of concept spaces on the ImageNet-1k binary judgment task. The results are as follows:

Comparison with one-hot vectors:

First, increasing dimensions through one-hot is not optimal from a computational performance perspective. The results in Reply Fig.2 demonstrated that a learnable 100-dim concept (the second bar from the left) can achieve better results (mean difference = 0.0043, 95% bootstrap CI [0.0021, 0.0056], 5000 resamples, two-sided permutation test with 10,000 permutations, $p = 0.0079$) than a 1000-dim one-hot (the third bar from the left). Second, one-hot has poor scalability. One-hot vectors are orthogonal, which means its dimension scales linearly with the number of classes. This is inefficient and biologically implausible for a large number of concepts. Therefore, in summary, whether from the perspective of overall performance or scalability, one-hot vectors are not a better choice.

Learnability of the concept space and CA/TS capacity are both crucial:

Trainable 20-dim concept space outperforms the frozen 20-dim random vectors (mean difference = 0.0192, 95% bootstrap CI [0.0185, 0.0200] with 5000 resamples, two-sided permutation test with 10,000 permutations, $p < 0.001$) and the frozen 20-dim Word2Vec vectors (mean difference = 0.0279, 95% bootstrap CI [0.0256, 0.0313] with 5000 resamples, two-sided permutation test with 10,000 permutations, $p < 0.001$). Indeed, given the frozen 20-dim random vectors, the CA/TS modules are re-organizing its weights to accommodate a fixed arbitrary space. When we reduced the number of CA/TS layers from 3 to 1, we can clearly see that the accuracy drops from 0.944 (the fourth bar from the left in Reply Fig.2) to 0.793 (the fifth bar from the left in Reply Fig.2), indicating that the current CA/TS with limited capacity is not sufficient to accurately complete the task in the fixed random concept space. However, if we permit concept vector learning in this setting (i.e., with one layer of the CA/TS module), the accuracy rises back to 0.954 (the last bar from the left in Reply Fig.2), which is close to the case of using 3 layers of CA/TS for 20-dim concept vectors (the first bar from the left in Reply Fig.2). This shows that in essence, the CATS Net requires the learnability of the concept space and the capacity of CA/TS to support it. When the CA/TS capacity is limited, the learnability of the concept space becomes crucial.

Reply Fig. 2. Ablation Studies on Concept Space Construction. The left most one bar was adopted from **initial manuscript Fig. 2a** (the Learned bar is with the selection of a pretrained ResNet50 as the feature extractor, a concept size of 20, 3-layer CA/TS modules, and two-phase training), while the others represent the average of mean accuracy across 5 independently initialized models after training, and each point represents the corresponding mean accuracy cross all categories.

We revised the relevant text in our manuscript:

(Results Section 2.1: line 219 - 247)

“Importantly, our empirical comparisons indicate that the learnability of both the concept vectors and the CA/TS modules is equally critical. We compared our approach with three alternative methods for constructing a fixed concept space (Fig S1b): using (1) frozen 20-dimensional random vectors, (2) frozen Word2Vec vectors projected to 20 dimensions, or (3) 1000-dimensional one-hot vectors.

The results revealed a crucial interplay between concept space learnability and network capacity. First, the trainable 20-dim space significantly outperforms its frozen counterparts (random: mean difference = 0.0192, 95% bootstrap CI [0.0185, 0.0200] with 5000 resamples, two-sided permutation test with 10,000 permutations, $p < 0.001$; Word2Vec: mean difference = 0.0279, 95% bootstrap CI [0.0256, 0.0313] with 5000 resamples, two-sided permutation test with 10,000 permutations, $p < 0.001$). This indicates that without learnability, the network is forced to merely reorganize its weights to accommodate a fixed, arbitrary space. This dynamic is magnified when network capacity is limited. For instance, when reducing the CA/TS modules from 3 layers to 1, performance with a frozen random space decreases from 0.944 to 0.793. However, simply allowing the concept vectors to become trainable in this low-capacity setting restores performance to 0.954. This demonstrates that the learnability of the concept space is not just beneficial but can be essential, compensating for limited network capacity. Second, we compared our method to the high-performing one-hot vector baseline. Our approach is superior from both a performance and scalability perspective. On performance, a learnable 100-dim concept space can achieve better results (mean difference = 0.0043, 95% bootstrap CI [0.0021, 0.0056], 5000 resamples, two-sided permutation test with 10,000 permutations, $p = 0.0079$) than the 1000-dim one-hot alternative. Furthermore, the one-hot representation has fundamental limitations for modeling cognition: its orthogonal nature precludes any semantic structure, and its dimensionality scales linearly with the number of classes---so adding concepts necessitates restructuring the entire space and re-estimating the mapping.”

2. On the Comparison with Word2Vec in the "Communication" Task:

We appreciate the observation regarding the Word2Vec-based concept transfer performance in the communication task. This study aims to demonstrate that our proposed mechanism is a viable model for human concept formation from sensorimotor stimulus, without the aid of human linguistic contents. We have shown that the CATS Net generated concepts achieve communication efficacy comparable to human linguistic concepts, thereby establishing feasibility rather than claiming superiority.

Learning Concepts from Stimulus: Our CATS Net learns its conceptual space from scratch, relying only on visual inputs and their corresponding labels, without prior exposure to language or human-defined semantic relationships. This provides a computational model for how conceptual knowledge might be acquired through direct perceptual experience, offering insights into fundamental learning processes that operate independently of linguistic frameworks.

Word2Vec as a "Human-Level" Benchmark: Word2Vec vectors, derived from vast amounts of human-generated text, represent well-established semantic structures. The fact that our self-generated concept space enables cross-network knowledge transfer at a level approaching that of language-derived representations demonstrates the validity of our proposed mechanism for concept formation.

Potential Benefits for Artificial Neural Networks:

Beyond validating human-like concept formation, our approach offers significant advantages for artificial neural networks. Not all concepts can be fully expressed through human language, and many concepts related to perception, abstraction, or specific domains may lack linguistic representation, such as concepts of reinforcement learning policy (see Appendix of this letter for details). Using CATS Net can enable artificial neural networks to autonomously generate appropriate concept representations for these situations, thereby breaking through the limitations of language-based concept spaces.

We revised the relevant text in our manuscript:

(*Discussion* Section 3: line 783 - 811)

"Modeling Visual Emerged Concepts vs. Applying Existing Knowledge. The preceding experiments, demonstrating success with both learned and human-derived concept spaces, highlight two distinct but complementary strengths of the CATS Net framework.

First, the new knowledge acquisition performance achieved using vectors from Word2Vec and the THINGS dataset validates the architectural flexibility of our dual-module system. It proves that the framework is not restricted to its endogenously formed concepts but can effectively interface with and leverage the rich structure of pre-existing, human-derived conceptual knowledge, whether sourced from language statistics or direct behavioral data. This compatibility is a crucial feature of the model's general applicability.

However, the primary contribution of our work lies not in merely utilizing these static, pre-existing spaces, but in modeling the process of concept formation itself. Unlike systems like Word2Vec, which distill knowledge from vast, human-generated text corpora, CATS Net demonstrates how a structured conceptual space can emerge *de novo* relying solely on visual inputs and task demands. This models a fundamental aspect of cognitive development: the ability to acquire concepts before, or independent of, language.

From this perspective, results of leave-one-out experiment using Word2Vec or THINGS SPOSE49 should be viewed not as a direct competitor, but as a benchmark representing a mature, culturally accumulated semantic system. The key insight is not whether our emerged concept space outperforms this benchmark in every task, but that it can organize a functionally effective conceptual structure that approaches the utility of human language-derived representations. The communication between two CATS Nets in the previous section, using their own generated concepts, is a demonstration to this capability. This validates our model as a plausible mechanism for the initial acquisition and application of conceptual knowledge, providing a potential computational understanding for how humans might build their conceptual world from perceptual experience.”

(Discussion Section 3: line 812 - 823)

“The CATS Net has the potentials for human-like knowledge formation and acquisition through other individuals. The first key advance is CATS Net's ability to develop functionally meaningful concept representations, unlike systems reliant on human-curated symbols (e.g., MLLMs). Our results indicate that meaningful abstraction can emerge from sensorimotor experience alone, offering a mechanistic analog to *symbol grounding* [36], where arbitrary signs acquire meaning through shared experiential basis. Not all concepts can be fully expressed through human language, and many concepts related to perception, abstraction, or specific domains may lack linguistic representation. Using CATS Net can enable artificial neural networks to generate appropriate concept representations for these situations (e.g., in none-visual domain such as reinforcement learning), thereby breaking through the limitations of human language-based concept spaces.”

Comment 4. I was very confused upon reading Method section 4.8 “Leave-one-out experiment and new concept abstraction”, because the description did not seem to correspond to any experiment or figure. Then I realized (in Method section 4.9) that some methodological tricks for data augmentation were used in some of the experiments (translation module) without any explanation in the main text.

Reply:

We thank you for pointing out this confusion regarding the presentation of the leave-one-out experiment. You are correct that the description in Method section 4.8 appears disconnected from the main experimental narrative. We acknowledge this is a writing clarity issue that needs to be addressed.

We revised the relevant text in our manuscript:

(Results Section 2.3: line 459 - 460)

“This translation module was trained with expanded concept vectors (Methods), to align the concept space between the two Nets.”

(Methods Section 4.8: all content in this section)

“This section describes the technical procedures underlying the communication experiment presented in Figure 4. The leave-one-out training and concept vector expansion serve two critical functions: (1) creating knowledge asymmetry between teacher and student networks, and (2) generating sufficient training data for the translation module that enables concept transfer.”

Comment 5. What was the dimensionality of the pretrained Word2vec representation? Was it 20, or did you

project it down to 20, and in this case, from what original dimension (300?), and what was the projection method (PCA)?

Reply:

We thank you for this important clarification question. The original pretrained Word2Vec representation had 300 dimensions, obtained from the fastText (**main text: Section 4.10**) library's pretrained English model (cc.en.300.bin). We then reduced these representations to 20 dimensions using fastText's built-in **reduce_model()** function, which employs Principal Component Analysis (PCA) as the dimensionality reduction method. We will clarify these technical details in the revised manuscript to ensure transparency about our Word2Vec implementation.

We revised the relevant text in our manuscript:

(*Method* Section 4.10: all content in this section)

“In these experiments, CATS Net was trained using the category-name word vectors as the predefined concept vector, which were provided by the fastText library. We used the pretrained 300-dimensional English word vectors (cc.en.300.bin) and reduced them to 20 dimensions using fastText's built-in *reduce_model()* function, which employs Principal Component Analysis (PCA) for dimensionality reduction. This 20-dimensional representation was chosen to match the dimensionality of our self-generated concept vectors, enabling direct comparison between the two concept spaces. The dataset was divided into two parts, D_{99} and D_1 , in the same way as in the leave-one-out concept abstraction experiment. CATS Net was directly trained by class label names, represented by their 20-dimensional Word2Vec embeddings, with images belonging to D_{99} . Then it was tested with the untrained class name corresponding to D_1 to identify the images. Experiments were also repeated 100 rounds with each category chosen as D_1 .”

Reviewer #2 (Remarks to the Author):

The manuscript titled “A neural network model for concept formation, understanding, and communication” by Guo et al. introduces CATS Net, a dual-module neural network architecture that combines concept abstraction and sensorimotor task-solving within a single computational framework. The authors show that this model not only creates low-dimensional concept representations but also restores sensorimotor functions through hierarchical gating, which aids in concept transfer and cross-model communication. They also discuss how the emergent representations of the model align with human neurosemantic spaces and brain response patterns in VOTC and semantic control regions. While this work offers an interesting approach by combining two types of established networks to explore new research directions, it is not clear what specific advancements this model provides over previous networks in terms of its goals and applications. Additionally, the assumptions of the model and other methodological details need further clarification.

Reply:

We appreciate your overall positive evaluation about our work and helpful comments from both AI and cognitive neuroscience sides. For the parts where you point out the goals, applications, and advancements of this study have not been clearly elaborated (also see general response), we have added new data, experiments, and arguments. For the details, please see below for the point-by-point response.

Major Points:

Comment 1. It seems the new model is designed for image classification with concept constraints. I had trouble fully appreciating the novelty of this creation. What is the difference between this creation and the classic ImageNet-based convolutional neural networks with attention modules? Will this network outperform existing networks in terms of image classification accuracy or computational resources? How is this network more advantageous than others? How is this network different and more advanced than the communication protocols used in multi-agent reinforcement learning? Without a clear comparison and explanation, I find it difficult to see the benefits and the rationale behind generating a new network by combining existing ones.

Reply:

We sincerely thank you for raising these critical questions. Under a classification-centric reading, it is natural to ask about performance advantages and comparisons with architectures such as ViT or attention-based CNNs. Instead, our work introduces a neuro-inspired computational framework designed to model a fundamental aspect of human intelligence: how abstract concepts are formed, understood, and communicated. The performance on classification tasks serves as a validation of our framework's *functional effectiveness*, rather than being the end goal itself. Also mentioned in the previous general response, our model is not limited to the visual modality (empirically demonstrated in the reply to your next Comment and Appendix in this letter) but is designed to understand the process of human concept learning.

1. Difference from Classic CNNs with Attention Modules:

The difference lies not in surpassing the accuracy of existing networks through incremental modifications, but in the core functional advantages that CNNs or ViTs cannot achieve.

The goal of CATS Net is to create a decoupled, low-dimensional, and *functionally meaningful* concept space that represents the network's *functionality*. This space is not merely for optimizing features; it serves to

compress an entire learned skill (e.g., the binary classification ability to identify "apples") into a compact vector. Each point in the concept space represents a functionality. The points corresponding to these different functionalities have a structured arrangement in the concept space. This structure remains highly similar across independently trained CATS Nets. Therefore, this similar structure can be utilized for cross-individual alignment, allowing the point of one functionality to be transferred to another individual, enabling it to acquire this functionality as well.

In stark contrast, classic attention modules in CNNs (e.g., SENet (Hu et al., 2018), CBAM (Woo et al., 2018)) or beyond CNNs (e.g., ViT (Dosovitskiy et al., 2021), multimodal cross-attention (Radford et al., 2021; Nagrani et al., 2022; Li et al., 2023)) are designed as intra-network feature optimization tools. Their primary function is to improve a single network's performance by dynamically re-weighting feature maps, enabling the model to focus on more salient channel-wise ("what") or spatial ("where") information. In essence, they are designed for enhancing an existing feature representation pipeline, but they do not involve the compression or reinstatement of network's *functionality*.

2. Difference from Communication Protocols in Multi-Agent Reinforcement Learning (MARL):

This is an excellent question that allows us to specify the nature of "communication" in our framework. The communication paradigm in CATS Net models knowledge acquisition and skill transfer, akin to human teaching. Agents are trained independently to develop their own unique sets of skills. Communication is the act of transmitting a fully formed, abstract concept from a "teacher" to a "student" (or from n agents to n agents), thereby endowing the student with a new skill it has not learned from experience (please kindly refer to our RL experiment in the reply to your next comment). This represents a mechanism for accumulating and sharing knowledge across a society of agents, rather than just coordinating their actions.

In contrast, communication in almost every MARL system is designed for coordination among agents working to solve a shared, immediate task. The communicated messages are typically about states, intentions, or actions for the common goal, and the agents are often trained jointly as a single system. The agent can hardly acquire new skill or functionality through communication, which is the advantage of CATS Net.

We hope this clarifies the positioning and the primary contributions of our work. We have revised the manuscript to make these distinctions clearer. Thank you again for your insightful feedback.

We revised the relevant text in our manuscript:

(*Introduction* Section 1: line 80 - 110)

"In humans, concept processing comprises two coupled capacities. First is concept formation: The higher-dimensional sensory-motor experience is compressed into lower-dimensional representational spaces [2–5], whose dimensionality typically ranges from 20 to several hundreds [6–10]. Second is concept understanding: Once the concepts are formed, they can be decoupled from immediate sensory input, being reactivated to reinstate sensorimotor states and flexibly combined [5, 11–15]. For example, hearing "last night's dinner" would elicit rich event-related imagery (Fig 1a), enabling communication of meanings through symbols. This bidirectional process is essential for concept formation, understanding, and communication in humans. A computational framework that simultaneously models both the concept formation and concept understanding remains a significant challenge in artificial intelligence and neuroscience.

Current paradigms fall short of integrating these two functions. On one hand, deep neural networks like ResNet and Vision Transformers [16, 17], or classic CNNs with attention modules [18, 19], excel at learning representations for classification. Mechanisms like attention, while powerful, serve as intra-network feature optimizers. However, a learned skill (e.g., the ability to identify a specific object among 1,000 objects) becomes entangled within millions of high-dimensional parameters. This skill (so called knowledge or functionality) is difficult to be compressed into a low-dimensional vector, making it hard to decouple from the network, activate without sensory input, or directly transfer to another agent. On the other hand, Multimodal Large Language Models (MLLMs) [20–22] demonstrate an impressive ability to align human texts with sensory representations, enabling cross-modal understanding, generation (e.g., text-to-image imagery of phrase “last night’s dinner”) and retrieval. Hub-and-spoke conceptual frameworks in cognitive neuroscience [5, 23] utilize a few to dozens of manually defined feature datasets to simulate the formation of concepts and the functional connectivity of cortex. Their limitation, however, is the reliance on pre-existing language symbols; they do not model the de novo formation of concepts directly from sensorimotor experience. What remains missing is a model that frames a concept as an executable, transferable function (i.e., a skill) that can be compressed into a low-dimensional vector and communicated, enabling the recipient to understand and acquire new capabilities without direct experience.”

Comment 2. This network is entirely vision-based, adding an extra question mark to the difference from well-trained and well-developed vision models. Is it possible for this new model to generalize to domains outside visual classification (e.g., auditory or embodied tasks), as these are also important for conceptual learning? Can this model be applied to temporally dynamic contexts?

Reply:

Yes, CATS Net is not specifically designed for visual concept extraction, and therefore can be fully generalized to other domains. We conducted a reinforcement learning (RL) multi-agent communication task. Each agent employs the CATS Net architecture for DQN learning. They independently explore different mazes and forms its own concept of RL policy. Then, different agents can directly communicate to acquire each other's concepts, navigating through unseen mazes. This confirms that CATS Net is not only designed for the visual domain, but also serves as a general-purpose architecture in temporally dynamic situations.

Specifically, we generated 3,800 9×9 perfect mazes (single-solution) via DFS; agents navigate from the bottom-left start to the top-right goal. A task-solving (TS) network was trained with DQN on a 6-dimensional input (x, y coordinates plus a one-hot action), outputting Q-values per action for the current state (Reply Fig. 3).

Reply Fig. 3. One example of the DFS generated maze (left) and its best policy decision matrix (right). All the generated mazes mark the bottom left as the starting point (x and y coordinates are -3 and -3) and the top right as the ending point (x and y coordinates are 3 and 3). The color block on the decision matrix, means that which action should be taken at this position in order to get to the ending point. For example, at the starting point, according to the Q-value calculated by the weights θ_{m1} of the TS Net, the agent should go up (marked as green).

Five CATS Net agents were trained: each learned 3,700 common mazes plus 20 unique mazes (3,720 total), with an 18-layer CA/TS (768 units per layer) and a 64-dimensional concept space; no translation module was used. During training, concept vectors for common mazes in agents 2–5 were aligned to agent 1’s via an L2 constraint. After training, each agent solved its seen mazes at near-perfect rates (agent 1: 3710 of 3720; agent 2: 3709 of 3720; agent 3: 3713 of 3720; agent 4: 3713 of 3720; agent 5: 3708 of 3720), and critically, could solve its 80 unseen mazes zero-shot by receiving the appropriate concept vector from another agent.

Using an ϵ -greedy policy ($\epsilon = 0.8$) and a 100-step threshold, zero-shot success rates on unseen mazes ranged from ~50% to 60% (Reply Fig. 4), far above chance (random policy success $\approx 0.25^{20} \approx 0$). These results indicate that CATS Net compresses complex task strategies into low-dimensional concept vectors and communicates executable skills without direct experience.

Reply Fig. 4. Communication results of unseen mazes for each CATS Net agent (from Agent 1 to Agent 5). All the agents can achieve a success rate far higher than the random policy.

Full settings and more details are provided in the Appendix. Given that the maze experiment lacks comparison with corresponding human brain data and behavioral data, and that this experiment is relatively independent of the content of the current work. **We prefer to avoid introducing this supplementary reinforcement learning experiment in the current manuscript due to the complexity, and instead focus on elaborating the computational mechanism of human concept learning raised by CATS Net.** These aspects will be reported in our next work.

Comment 3. The idea that concept vectors can be communicated between independently trained networks is interesting. However, the manuscript could explain whether the translation module maintains semantic detail. Is any information lost during translation, especially when concepts involve subtle or overlapping features?

Reply:

We thank you for this insightful question about semantic detail preservation in the translation module. To

address this concern, we conducted representational similarity analysis (RSA) to probe the layer-wise activations of the translation module, processed across 100 teacher-student network pairs.

The translation module is a multiple-layer perceptron (MLP), with ten hidden layers containing 500 neurons each. Given all teacher's concept vectors as input, through visualization of the layer-wise RDMs, it is expected to see that the overall patterns are reserved while some certain variations in local pattern details. Also, the quantitative correlation analysis reveals that the correlation between the RDMs of each layer and the RDM of input data decreases during the information transmission process, indicating that there is indeed some loss of information. See below for details:

1. Representative Case Study.

To illustrate the translation process, we present a detailed analysis of one teacher-student pair (randomly chosen, the "apple" category was withheld from the student). For each translation module, we extracted feature vectors from the input layer, all 11 ReLU hidden layers, and the output layer when processing the teacher's 100 concept vectors (Reply Fig. 5a).

Qualitatively, we found that semantic clustering patterns observed in RDM remained consistent across layers, demonstrating that those concepts involve subtle or overlapping features (e.g., the top left cluster) almost maintain their proximity throughout the translation process. Quantitatively, the layer-wise RDM correlation analysis for this specific translation module reveals how semantic relationships evolve through the network: starting with high input to intermediate-layer correlations (0.930 between input and relu_0), the correlations gradually decrease through successive layers (e.g., input with relu_5: 0.400, input with relu_10: 0.344), demonstrating systematic information processing rather than arbitrary loss (the first column in Reply Fig. 5b).

Reply Fig. 5. Layer-wise correlation matrix. a, for this translation module, given all 100 teacher concept vectors as input, we recorded the layer-wise activations and conducted layer-wise RDM (Pearson’s correlation). **b**, we computed the layer-wise RDM Spearman’s correlation similarity matrix. **c**, the average layer-wise RDM Spearman’s correlation similarity across all 100 translation modules. **d**, One-sample *t*-test of each value at translation module group level.

2. Statistical Evidence Across All 100 Translation Modules:

Extending this analysis to all 100 teacher-student pairs, we computed RDM correlations between all layer pairs across translation modules. The aggregated results (the first column in Reply Fig. 5c) show that correlations between input and successive layers range from 0.93 (input - relu_0) to 0.29 (input - output), with all correlations being highly significant ($p < 0.001$, Reply Fig. 5d). Adjacent layers maintain very high correlations (> 0.83 , the secondary diagonal in Reply Fig. 5c), indicating gradual and controlled information transformation rather than abrupt semantic loss.

We revised the relevant text in our manuscript:

(Results Section 2.3: line 460 - 508)

“To investigate whether the translation module preserves semantic details during concept transfer, we

analyzed the internal representations across all layers of the translation module. Using visualization of layer-wise RDM (Fig 4d, Fig S4a&b), we found that semantic clustering patterns observed in RDM remained consistent across layers, demonstrating that conceptually related categories (e.g., the top left cluster) maintain their proximity throughout the translation process. Through RDM correlation analysis across 100 teacher-student pairs (Fig S4c), we found that the translation module systematically preserves semantic relationships while performing functional adaptation (Fig S4d for statistical significance). Specifically, RDM correlations between input and successive layers showed a gradual decrease (from 0.93 to 0.29 at the output layer, all $p < 0.001$), indicating controlled information compression rather than arbitrary loss.”

Comment 4. The authors used Word2Vec to summarize the human concept space. While Word2Vec is a powerful tool for exploring the human concept space, the concept range in this study is limited and focused only on concrete concepts. It would be better if real human-generated data could be considered, and the authors should discuss how the representations differ between concrete and nonconcrete concepts.

Reply:

We thank you for this important comment regarding the use of Word2Vec and the need for real human-generated data. We added the experiments on real human-generated data and would like to clarify several key points:

1. Real Human-Generated Data Validation:

Following your suggestion, we have conducted additional experiments using real human-generated concept representations from the SPOSE49 model (Hebart et al., 2020), which generates 49 replicable and semantically meaningful dimensions in a data-driven manner based on large-scale images similarity judgment human data across 1,854 objects (THINGS dataset, Hebart et al., 2019), providing genuine human-derived conceptual representations rather than text-derived embeddings. We performed the same leave-one-out transfer experiments using the original images from THINGS and these produced 49-dimensional vectors instead of Word2Vec embeddings. The results (mean accuracy: 69.67%, std: 15.82%, threshold 0.5, $t(99)=12.43$, one-tail $p < 0.001$, Cohen's $d = 1.24$, 95% CI [0.6653, 0.7281]) demonstrate that our CATS Net architecture is fully compatible with real human-generated conceptual data and achieves comparable performance to Word2Vec-based experiments. This validates that our model can work with authentic human conceptual representations, not just linguistic embeddings.

Reply Fig. 6. Performance on unseen concepts under the leave-one-out paradigm using THINGS and 49-dim human-generated embedding vectors. We randomly chose 100 categories from 1,854 object categories from THINGS as 100 runs of leave-one-out experiment. Each point in the figure represents one category chosen for leave-one-out experiment.

We revised the relevant text in our manuscript:

(Results Section 2.4: line 526 - 528)

“To test whether our CATS Net is able to utilize the concepts generated by humans, we evaluated its performance using both language-derived and direct human behavioral-derived concept spaces.”

(Results Section 2.4: line 575 - 586)

“**Human Behavioral Data Validation.** To further validate our architecture's compatibility with human behavioral-derived concept spaces, we conducted experiments using SPOSE49 model. Using these human-generated concept vectors, we replicated the leave-one-out experimental paradigm described above. The results demonstrate that our CATS Net architecture achieves comparable performance when configured with human behavioral data (mean accuracy (SD) 0.6967 (0.1582), threshold 0.5, $t(99)=12.43$, one-tail $p < 0.001$, Cohen's $d = 1.24$, 95% CI [0.6653, 0.7281]), confirming that our dual-module framework can effectively exploit not only language-derived concept spaces but also genuine human perceptual and conceptual structures. This validation strengthens the claim that our architecture provides a general computational framework for concept formation and understanding that is compatible with authentic human cognitive processes.”

2. Representational Differences Between Concrete and Nonconcrete Concepts:

We agree that the discussion about the representational differences between concrete and abstract concepts would help improve our research. Our current research primarily focused on concrete concepts because, in the visual domain, abstract concepts lack of bounded, identifiable and clearly perceivable referent (i.e., an image in visual domain; Borghi et al., 2017), and show higher inter- and intra-individual variability than concrete ones (Barsalou, 2003). Accordingly, current abstract-label image datasets face fuzzy category boundaries and limited annotation reliability; recent attempts report only modest improvement and substantial between-label confusions (e.g., Martinez Pandiani et al., 2023; see discussion in Pandiani & Presutti, 2024). For this reason, our analyses were scoped to concrete categories. Even so, we extracted abstract-domain features from Binder65 models on ImageNet-1k dataset, and revealed the significant correspondence between

CATS concept layer and Binder65 abstract domains (e.g., spatial, social, emotional; see Fig. S3), suggesting capacity relevant to processing abstract concepts, despite not optimized for abstract-label recognition. In addition, our extended navigation study demonstrated cross-task generalization, which we view as a practical route to investigating abstract concepts beyond purely visual categorization.

We revised the relevant text in our manuscript:

(Discussion Section 3: line 871 - 884)

“Our research focused primarily on concrete concepts, reflecting inherent challenges in representing nonconcrete concepts in vision. Nonconcrete concepts lack bounded, identifiable referents [40] and exhibit greater inter- and intra-individual variability than concrete concepts [41]. In line with this, current image datasets for nonconcrete concepts have fuzzy category boundaries and limited annotation reliability; recent attempts report only modest improvement and substantial between-label confusion [42, 43]. On this basis, we restricted our analyses to concrete categories. Even so, exploratory analyses revealed significant correspondence between CATS concept-layer representations and Binder65 abstract domains (e.g., spatial, social, emotional; Fig. S3), indicating capacity relevant to abstract-concept processing despite the models not being explicitly optimized for abstract-label recognition. Future work could extend this framework to other modalities to systematically test CATS’s ability to instantiate abstract conceptual spaces.”

Comment 5. The matrices in Figure 4a look very different from each other, and the correlations are low. Is the significance mainly caused by the large number of features?

Reply:

We thank you for these careful observations and important questions. We address the visual differences in Figure 4a and the statistical significance concerns separately:

1. Visual Differences Between Matrices

The apparent visual differences between matrices in **initial manuscript Fig. 4a** (now in **revised manuscript Fig. 5a**) primarily stemmed from limitations in the original plotting configuration. To enhance comparability, we have re-optimized our visualization approach by implementing clustering analysis prior to plotting and converting values to percentage representations, making the structural similarities between matrices more intuitive. The updated results are presented in the **Reply Fig. 7** and **revised manuscript Fig. 5a**, clearly demonstrating the structural correspondence between the two representational spaces.

Reply Fig. 7. Clustered visualization of concept RDM and Word2Vec RDM. RDMs of learned concept vectors (left)

versus Word2Vec vectors (right). We use hierarchical clustering to rearrange the index and convert the values to percentage representations for better visualization.

2. Correlation Magnitude and Statistical Significance

We agree that naïve parametric tests can overstate significance by treating RDM cells as independent. Accordingly, we used a Mantel test (synchronous label permutation, 10,000 permutations) which preserves the dependence structure: $\rho = 0.24$, $p < 0.001$. We also report distribution-free uncertainty via bootstrap resampling (5,000 resamples), yielding 95% CI [0.154, 0.366]. These results indicate that significance is not only driven by the large number of features but reflects a genuine effect.

We revised the relevant text in our manuscript:

(Results Section 2.4: line 570 - 574)

“Although the Word2Vec vectors are derived from word co-occurrence statistics in large text corpora, which is fundamentally different from our model, their RDMs still show a significant correlation with ours (Spearman $\rho = 0.24$; Mantel $p < 0.001$, 10,000 permutations; bootstrap 95% CI [0.154, 0.366], 5,000 resamples; Fig. 5a).”

Comment 6. The model-brain correlations are low too (< 0.1). What do the noise ceilings look like across the brain? Do the correlations mainly reflect differences in noise ceilings? How robust are these correlations across varying model initializations, dataset splits, or semantic ontologies? Were multiple neurosemantic models tested, and how does Binder65 compare to alternatives like WordNet-based semantic spaces?

Reply:

We thank you for these insightful observations and important questions.

1. Noise Ceiling Analysis

We agree that the absolute magnitudes of the model–brain correlation are low, but this primarily reflects the inherently low signal-to-noise ratio of fMRI data and the functional complexity of VOTC.

Low signal-to-noise ratio of fMRI data.

To provide the noise ceiling aligned with our statistic (per-subject correlation \rightarrow Fisher-z \rightarrow subject-mean), we estimated subject-wise reliabilities (leave-one-out against the group mean, corrected by the group split-half reliability; Nili et al., 2014), and model-wise reliability (leave-one-out across the 30 model instances), and then combined them in the z-domain to obtain the noise ceiling:

$$NC_z = \frac{1}{S} \sum_S \operatorname{atanh}(\sqrt{rel_s \cdot rel_{model}})$$

The resulting ceiling were: VOTC–concept, $NC_z = 0.25$; Multiple-Demand network–CA1, $NC_z = 0.27$; and Semantic-Control network–CA1, $NC_z = 0.24$. Relative to these ceilings, the effects are not low (e.g., concept–VOTC mean Fisher-z of 0.04 corresponds to 16% of the ceiling). Crucially, the correspondence replicated across model instances and subjects: across 30 instances, the group-mean Fisher-z was reliably above zero (concept–VOTC: $t(29) = 9.27$, $p < 0.001$, Cohen’s $d = 1.70$), inconsistent with a noise-only explanation.

The functional complexity of VOTC.

VOTC supports multiple computational processes spanning different hierarchical levels, including low-level

visual feature extraction, high-level visual-semantic representations, and potentially multimodal semantic integration (see discussion in Calzavarini, 2024; Bi et al., 2016). Our concept layer targeted the latter functions but not low-level visual features, which constrained its attainable ceiling in VOTC. Consistent with this scope, a conceptual but non-visual model (SPOSE49) reached a similar VOTC correspondence (Fisher-z $\rho = 0.056$; mean across 26 participants). For comparison, a widely used baseline (ResNet-50 average pool layer, which is also the sensory input layer of our model), also yielded low VOTC correspondence (mean Fisher-z transformed $\rho = 0.007$) in our dataset.

Taken together, the noise ceiling and the functional complexity of VOTC jointly constraint the attainable model–brain correlation; baseline results aligned with this limited upper bound, and instance-group-level analyses indicated the effect was not purely attributable to noise.

We revised the relevant text in our manuscript:

(Results Section 2.5: line 693 - 698)

“Across ROIs, effect size was reliable across 30 independently initialized models (one-sample tests on Fisher-z means) and should be interpreted relative to the noise ceilings (VOTC–concept $NC_z = 0.25$; Multiple-Demand–CA1 $NC_z = 0.27$; Semantic-Control–CA1 $NC_z = 0.24$; see Methods). For context, the VOTC correspondence of a widely used baseline model (ResNet-50) and SPOSE49 model were 0.007 and 0.056, respectively (mean Fisher-z transformed ρ , mean across 26 participants).”

(Methods Section 4.17: line 1289 - 1316)

“Our primary effect size for each model instance i is computed by correlating the instance’s RDM with each participant’s RDM within a ROI using Spearman’s ρ , applying a Fisher-z transform, and averaging across participants. So, to provide an upper bound that is commensurate with this statistic, we estimate a noise ceiling (NC) in the z-domain that jointly reflects measurement reliability on the participant side and stochastic variability on the model-instance side.

For Participant-wise reliability rel_s , we first estimate the group-mean RDM reliability rel_{group} via participant split-half half-sample means correlated across many random 50/50 splits, Fisher-z averaged and Spearman-Brown corrected), then relate each participant to the leave-one-out group mean, $r(X_s, \overline{X_{-s}})$, and obtain:

$$rel_s \geq \frac{r^2(X_s, \overline{X_{-s}})}{rel_{group}}$$

For single-instance model reliability rel_{model} , we estimate the reliability of a single model instance using a leave-one-out (LOO) approximation: for each instance i , we compute the correlation between M_i , and the mean RDM of the remaining $M - 1$ instances, $r_i = \rho(M_i, \overline{M_{-i}})$; we then average r_i in the Fisher-z domain and back-transform to obtain rel_{model} .

Finally, the expected correlation between a single participant and a single instance is bounded by

$\sqrt{rel_s rel_{model}}$. Because our effect averages Fisher-z values across participants, we aggregate the bound in the same domain:

$$NC_z = \frac{1}{S} \sum_{s=1}^S \operatorname{atanh}(\sqrt{rel_s rel_{model}})$$

”

2. Robustness Analysis

Model Initialization.

We independently trained 30 models and obtained highly consistent correlation results at the group level using Fisher-z transformed, ceiling-corrected correlations: **concept–VOTC**, $t(29) = 9.27$, $p < 0.001$, Cohen’s $d = 1.70$; **CA1–Semantic Control**, $t(29) = 6.44$, $p < 0.001$, Cohen’s $d = 1.18$).

Datasets Splits.

According to your suggestion, we computed the instance-average model-brain correspondence per participant (averaging across the 30 independently initialized models). In VOTC, the per-participant correspondences were all > 0 (mean $\rho \pm SE = 0.095 \pm 0.009$), and were significantly greater than zero at the group level ($t(25) = 10.90$, $p < 0.001$, Cohen’s $d = 2.14$), indicating that the CATS concept layer can stably correspond to human VOTC across subjects.

In the semantic control network, we similarly observed consistent positive correlations between the CA1 layer of 30 models and subjects’ neural representations ($t(25) = 3.51$, $p < 0.001$, Cohen’s $d = 0.69$). This further supports the correspondence between the control module in the CATS framework and the semantic control network. In addition, the CA1 layer of all thirty models also demonstrated significant fitting advantages for the Semantic Control Network in each subject, relative to the Multiple Demand Network ($t(25) = 2.23$, $p = 0.035$, Cohen’s $d = 0.43$).

Semantic Ontologies.

Since WT95 contains three classic human semantic categories (animals, large non-manipulable objects, and small manipulable objects), we divided it into three subsets and conducted analyses within each subset following the same RSA-searchlight analysis procedure. Specifically, we generated neural RDMs based on stimuli within each subset and compared them with feature RDMs from the model’s concept layer. Given that previous whole-brain analyses already demonstrated high correspondence between the concept layer and human VOTC, this analysis was confined within a predefined VOTC mask. Results showed that in each semantic category, CATS models captured significant neural effects within VOTC and demonstrated unique advantages relative to sensory input layer representations (ResNet-50 output) (voxel-level $p < 0.001$, one-tailed; cluster-level FWE-corrected $p < 0.05$; see **Reply Fig. 8** and **revised manuscript Fig. S8**).

Specifically, for the animal category, our analysis identified significant clusters in bilateral fusiform gyrus (FG), extending to the lateral occipital complex (LOC). For large non-manipulable objects, we detected three spatially relatively independent significant clusters: one located in the left occipital pole (OP), another in the right occipital pole extending anteriorly to the lingual gyrus (LING) and medial FG, and a third in the left medial FG. Finally, for small manipulable objects, we identified a significant cluster located in the left FG. The weak effect may come from dataset composition—few representative tool exemplars and marked within-category heterogeneity (spanning small, rounded objects, e.g., buttons, to elongated objects, e.g., fishing rods).

Reply Fig. 8. Searchlight RSA results within the VOTC mask for three common semantic categories (animals, large non-manipulable objects, small manipulable objects). The maps show t-values reflecting the correspondence between concept layer representations and brain activity. Results are thresholded at voxel-level $p < 0.001$ (one-tailed) and cluster-level family-wise error (FWE) corrected $p < 0.05$.

This analysis has been added to the revised manuscript:

(Results Section 2.5: line 708 - 711)

“These findings aligned with our ROI results, confirming that the concept layer corresponds to neural representations in VOTC, while the CA module predominantly corresponds to activities in semantic control regions (for validation, see Supplementary Information).”

3. Multiple Neurosemantic Model Testing

We additionally considered another neurosemantic model—the SPOSE49 model (Hebart et al., 2020), which generates 49 replicable and semantically meaningful dimensions in a data-driven manner based on large-scale concept images similarity judgment data. We matched 334 overlapping concepts between ImageNet-1k and THINGS. And then we performed RSA analyses at both the group and model-instance levels. At the group level, the mean concept-layer RDM (averaged across 30 instances) correlated with the SPOSE49 RDM (Spearman’s $\rho = 0.29, p < 0.001$); at the instance level, Spearman correlations were Fisher-z transformed and were significantly greater than zero across instances ($t(29) = 25.60, p < 0.001$, Cohen’s $d = 4.75$), further supporting that CATS concept space aligned with human semantic spaces.

Reply Fig. 9. The heatmap of the averaged CATS RDM, SPOSE49 RDMs and Binder65 RDM.

This analysis has been added to the revised manuscript:

(Results Section 2.2: line 324 - 358)

“We further examine the extent to which the conceptual space in the CATS Net models is similar to the human conceptual space more directly. To this end, we adopted two complementary human semantic models. The first is a neurocognitive semantic model, which posited 65 specific dimensions that are neurobiologically grounded (Binder et al., 2016) (hereafter referred to as Binder65). The second is a behaviour-derived semantic model, which generated 49 meaningful dimensions through data-driven methods applied to large-scale human concept images similarity judgement data from the THINGS dataset (Hebart et al., 2020) (hereafter referred to as SPOSE49).

We employed representational similarity analysis (RSA) (Kriegeskorte et al., 2008) which allows to compare second-order isomorphisms between different systems. First, we identified 334 concepts shared between ImageNet-1k and the THINGS dataset. Features of each ImageNet-1k concept were then extracted from both our CATS concept layer and the semantic model spaces. Two concepts could not be represented in the Binder65 feature space, resulting a final set of 332 concepts for subsequent analyses. Representational dissimilarity matrices (RDMs) were constructed by calculating pairwise Pearson’s distance between concept feature vectors for each model.

We assessed correspondence at both group and instance levels. At the group level, we averaged RDMs across the 30 CATS instances and correlated this average CATS RDM with the Binder65-derived and SPOSE49-derived RDMs. At the instance level, we correlated each individual CATS instance RDM with the Binder65-derived and SPOSE49-derived RDMs, then applied one-sample *t*-tests on Fisher-*z*-transformed correlations to determine whether group-level correspondences significantly exceeded zero.

As shown in Fig 3a, the average CATS Concept RDM showed significant and moderate correlations with Binder65 RDM (Spearman’s $\rho = 0.14$, $p < 0.001$) and SPOSE49 RDM (Spearman’s $\rho = 0.29$, $p < 0.001$). For the instance-level, the analyses revealed similar patterns: Binder65, $t(29) = 18.28$, $p < 0.001$, Cohen’s $d = 3.39$, 95% CI [0.03, 0.04]; SPOSE49, $t(29) = 25.43$, $p < 0.001$, Cohen’s $d = 4.72$, 95% CI [0.06, 0.07]. Taken together, these findings provide convergent evidence that our CATS Net, despite being trained solely on visual categorization tasks, generates a concept space structurally similar to human conceptual organization. Notably, this similarity seems to be able to reflect CATS’s ability to capture abstract dimensions (especially considering the abstract information explicitly implemented in the Binder65), as further evidenced by the significant correspondence with nonvisual dimensions of Binder65 (e.g., spatial, temporal, emotional; see Figure S2).”

3. Comparison of Binder65 to WordNet-Based Semantic Spaces

The Binder65 model differs from traditional WordNet-based semantic spaces. Binder65 dimensions derive from neurocognitive literature reviews, with large-scale subject ratings of concepts along these dimensions, making them more closely aligned with neural representations. In contrast, WordNet was manually constructed by experts, emphasizing hierarchical relationships between words rather than dimensional representations, and lacks support from large-scale behavioral data, thus having limitations in applicability and compatibility with visual tasks. Although WordNet avoids the arbitrariness of dimensional design, its structure primarily reflects linguistic relationships rather than perceptual or cognitive dimensions. In comparison, SPOSE49 combined advantages in this regard: it was generated based on large-scale images similarity judgments, avoiding the subjectivity of artificial dimensional design and being more compatible with visual classification task contexts.

Comment 7. While CAM-based visualizations are shown, providing more details on the interpretability of the 20-dimensional concept space would be helpful. Are there prominent dimensions that can be labeled semantically (e.g., animacy, shape, size)?

Reply:

We thank you for this important suggestion regarding the interpretability of the concept space. We acknowledge the significance of providing semantic labels for the concept space dimensions in CATS models, and have conducted relevant analyses.

To systematically decode the semantic structure of the CATS concept space, we employed the SPOSE49 model as a reference framework. SPOSE49 generates 49 concept space dimensions in a data-driven manner based on large-scale object similarity judgment data, and has been validated to effectively explain human similarity judgment behaviors (Hebart et al., 2020) and neural response patterns (Contier et al., 2024).

For each SPOSE49 dimension we adopted a best-match procedure: within every independently initialized CATS instance, we computed Pearson correlations between that SPOSE dimension and all CATS dimensions and retained the maximum as the “best-match” score for that instance. This yielded, for each SPOSE dimension, a distribution of 30 best-match correlations across CATS instances. **Reply Fig. 10** highlights four SPOSE dimensions—metal/tool, food, furniture, and long/thin—for which almost all CATS instances exceeded the nominal significance threshold ($p < 0.05$; $n = 334$ concepts), indicating convergence on similar semantic structure despite different initializations. The full set of results across all 49 SPOSE dimensions is presented in **revised manuscript Fig. S3**.

Reply Fig. 10. Consistency of SPOSE–CATS best-match correlations across model instances. Each polar “rose” chart corresponds to one SPOSE49 semantic dimension (shown: metal/tool, food, furniture, and long/thin). For each of the 30 independently initialized CATS models, we identified the CATS dimension that maximized the Pearson correlation with the given SPOSE dimension; the height of each petal encodes that best-match correlation (r) for one instance. The red circle marks the nominal significance threshold ($p < 0.05$ for $n = 334$ concepts). Insets report the mean and maximum correlations across instances, and the count of instances above threshold. Results for all 49 SPOSE dimensions are provided in **revised manuscript Fig. S3** (you could also see the figure in ZIP file for details).

This analysis has been added to the revised manuscript:

(Results Section 2.2: line 361 - 406)

“To further explore the interpretability of our CATS concept space, we tried to provide semantic labels for

our concept space dimensions. We employed the SPOSE49 model as a reference framework, because it captures finer-grained, visually relevant features that have been validated to effectively explain human similarity judgment behaviors (Hebart et al., 2020) and neural response patterns (Contier et al., 2024).

Specifically, we adopted a best-match procedure for each SPOSE49 dimension. Within each CATS instance, we computed Pearson correlations between a given SPOSE dimension and all CATS dimensions, retaining the maximum correlation as the “best-match” score for that instance. This yielded, for each SPOSE dimension, a distribution of 30 best-match correlations across CATS instances. Figure 3b highlighted four SPOSE dimensions (metal/tool, food, furniture, and long/thin) for which almost all 30 CATS instances exceeded the nominal significance threshold ($p < 0.05$; $n = 334$ concepts), indicating robust convergence on similar semantic structure despite different random initializations. The complete results across all 49 SPOSE dimensions were presented in Figure S3.”

Comment 8. The authors note that the CA module aligns with the semantic control network. Is there any previous research connecting multiplicative gating (as used here) to control regions such as IFG or dmPFC?

Reply:

We thank you for this important theoretical question. Regarding the connection between multiplicative gating mechanisms and semantic control networks, as far as we know, there are few studies researching about the computational mechanism of cortical control regions, especially the semantic control network. The existing literature primarily focuses on two relatively independent research directions: first, delineating functional boundaries between semantic control networks and other control networks through different experimental paradigms (Diachek et al., 2020; Jefferies & Lambon Ralph, 2006; Wang et al., 2024); second, examining dynamic interaction patterns between control and representational regions using connectivity analysis methods (Chiou et al., 2018; Evans et al., 2020). However, these studies have not attempted explicit theoretical modeling of the control computational mechanisms themselves.

While a few connectionists computational frameworks (such as Hoffman et al., 2018) do introduce multiplicative gating forms to explain control processes in semantic cognition, these models are primarily limited to replicating behavioral patterns in specific patient populations and lack direct comparative validation with neural representations of brain semantic control networks.

Therefore, our work fills this gap. The CATS framework we propose explicitly introduces a multiplicative gating-based control module within the sensory task framework, and leverages the representational capabilities of large-scale computational models to enable the direct comparison between such gating control representations and neural representations of brain semantic control networks. It should be noted that we have not systematically compared different gating mechanisms in fitting semantic control networks, thus we cannot definitively assert that multiplicative gating represents the exact computational mechanism employed by these networks. However, our results demonstrate that multiplicative gating serves as a reasonable and effective computational candidate for semantic control networks.

We have also supplemented relevant discussion in the revised manuscript:

(Discussion Section 3: line 851 - 863)

“Critically, the CA module demonstrates significant correspondence with the semantic control network, highlighting its functional resemblance to frontoparietal systems that flexibly gate conceptual knowledge. This alignment suggests that our model captures mechanisms specific to semantic control processing, whose computational principles remain largely unexplored in the current literature. We propose that the semantic control network may implement a task-related feature gating mechanism similar to our CA module, selectively amplifying or inhibiting semantic features (the inputs from the sensory layer) according to task demands (in our study, different category judgment tasks). This dissociation—concept representation in sensory regions and concept control in higher-order networks—echoes recent neurocognitive models positing distributed yet integrated semantic architectures [5]. Future studies could extend this framework to other modalities to systematically investigate the ability of CATS to generate abstract conceptual spaces.”

Minor Points:

Comment 9. The paper makes strong claims about “human-like” communication. While partly justified, the authors might consider softening this claim or clarifying that “symbolic transfer” does not yet equal natural language communication.

Reply:

We thank you for this point regarding our characterization of "human-like communication." We replaced the expression of "human-like communication" with "concept-based communication", "conceptual communication", etc.

We have revised our manuscript to softening this claim in the manuscript:

(Discussion Section 3: line 829 - 833)

“This represents an early form of conceptual communication that captures foundational principles of how humans use symbols to share conceptual knowledge, though it currently operates as a simplified abstraction that does not encompass the full complexity of natural language communication.”

Reviewer #2 (Remarks on code availability):

I reviewed the code provided by the authors. The repository is well-documented and clearly organized, with modular and readable code that closely aligns with the methodological descriptions in the paper.

The repository includes a comprehensive README file, which outlines the installation steps, dependencies, and usage instructions in a straightforward manner. From inspecting the code and the structure of the provided scripts and configuration files, it is clear that the authors have made a serious effort to ensure usability and reproducibility. Although I did not execute the code, the structure and documentation suggest that the results should be reproducible with the provided instructions and standard computational resources.

Overall, I think the codebase is a valuable resource for the community and should allow researchers to validate, reuse, and build upon the authors’ proposed CATS Net architecture.

Reply to code availability:

We are very grateful for your praise. We believe that making all code implementation details open source will

help future research in both AI and cognitive neuroscience communities.

In summary, we thank reviewers again for your constructive comments and kindly help.

Sincerely,

Shan Yu

Laboratory of Brain Atlas and Brain-inspired Intelligence

School of Artificial Intelligence, University of Chinese Academy of Sciences

Institute of Automation, Chinese Academy of Sciences

Beijing, China

shan.yu@nlpr.ia.ac.cn

Yanchao Bi

Boya Professor,

School of Psychological and Cognitive Sciences

IDG/McGovern Institute for Brain Research

Institute for Artificial Intelligence

Peking University, China

ybi@pku.edu.cn

Appendix: Details of CATS Net on reinforcement learning task

CATS Net is not specifically designed for visual concept extraction, and therefore can be fully generalized to other domains. The key lies on the ability of the CATS Net to compress high-dimensional functionally meaningful network parameter θ_i to low-dimensional concept vector c_i . To demonstrate the model's capability in a domain where the functionality tied with concepts c_i are not pre-defined by human, we have conducted a new experiment in the Reinforcement Learning (RL) domain, which is a multi-agent multi-maze-solving and communication task. In this task, different CATS Nets learn to abstract different maze-solving strategies into concept vectors. These concept vectors can then be communicated between different CATS Net agents to transfer maze-navigation skills.

A. Experiment settings

Specifically, we used the Depth-First Search (DFS) algorithm to generate 3,800 9x9 perfect mazes. These mazes have no loops or cycles, meaning there is only one path between any two positions. This is extremely challenging for reinforcement learning agents because of the only one solution out. The left panel of **Reply Fig. 3.** below shows one of these mazes.

Considering for a given maze, the goal of the agent is to find a path to the ending position (top right, marked as yellow) and we can use the traditional DQN approach for reinforcement learning. Let's start with a TS Net. This TS Net f_θ adopts a 6-dimensional real number vector as input (2 dimensions represent the agent's x and y coordinates in the maze, and the remaining 4 dimensions use a one-hot encoding to represent 4 actions: up, down, left, and right). Its output is a 1-dimensional real number, representing the Q-value of taking a certain action at a certain location. By training the TS Net using the Bellman equation, we can obtain a set of TS Net weights θ_{m1} specific to the maze 1. Same as the description before, this weight vector θ_{m1} resides in a high-dimensional parameter space and has as many dimensions as TS Net has parameters. This set of weights θ_{m1} , represents the *maze-solving functionality* of this specific maze, because it contains the information about which action should be adopted at each position. For the example maze shown in the right panel of **Reply Fig. 3.**, we mark the best policy decision matrix on the right side.

Naturally, for complex functional behaviors like maze navigation policy, there is no predefined policy-level concept space like the simple Word2Vec (which is also a concern in **Reviewer #1, Comment 3**). To solve different mazes, we can only replace a new set of TS Net weights $\{\theta_{m1}, \theta_{m2}, \theta_{m3}, \dots\}$ to obtain a new maze-solving policy. Just like described above, our CATS Net can naturally handle this situation by compress these extremely high-dimensional discriminative network parameters $\{\theta_{m1}, \theta_{m2}, \theta_{m3}, \dots\}$ into a low-dimensional concept vector space $\{c_{m1}, c_{m2}, c_{m3}, \dots\}$. Also, this compression is achieved through learnable concept vectors and the CA/TS modules store some common information for maze navigation. In summary, we can use a single CATS Net to learn multiple mazes in the *reinforcement learning task*, with a functionally meaningful concept space emerged.

It's worth noting that the concept vector we've compressed here doesn't just correspond to discrete actions like up, down, left, and right. Instead, it includes complex information such as the strategy for solving the current maze. Strictly speaking, it's more appropriate to call it a strategy vector. But for the sake of consistency, we'll still refer to it as a concept vector here.

B. Communication paradigm and results

Given a clear explanation above, we shall step into the communication paradigm in of this experiment. We have set up a total of 5 CATS Nets. As mentioned earlier, a total of 3800 different mazes has been generated. Here, we divide the mazes into 3700:20:20:20:20. All 5 CATS Nets learn the common 3700 mazes, and then each Net also receive 20 unique mazes that it exclusively learns. Namely, each agent learns 3720 mazes in total, while 80 unique mazes are left for testing. For each agent, its unseen 80 mazes are learned by one of the remaining four agents. The purpose of the entire communication experiment is to enable an agent to directly navigate out of a maze it has never seen before, solely through concept communication, without the need to learn strategies in the maze itself (**Reply Fig. 11**).

Reply Fig. 11. Communication paradigm of CATS Net on reinforcement learning domain. All the agents will learn a set of common mazes, while leaving some unseen mazes. The unseen mazes for each agent will be acquired by one of the other agents. The agent will receive concept vectors from other agents to successfully solve these unseen ones for itself.

Unlike the pre-trained ResNet feature extractor required in the visual domain, for each CATS Net in this experiment, we set up an 18-layer CA/TS with 768 neurons each layer, which is a two-way fully connected network that accepts a 6-dimensional input and produces a 1-dimensional output. The concept size is set to 64, with each agent randomly initialized with 3720 concepts. Furthermore, we removed the translation module used in the visual domain. During training, we first train the 1st agent using Bellman equation. After training is completed, the first agent generates its own concept vectors for the common maze. When training the 2nd to 5th agents, besides the Bellman equation for RL training goal as the 1st agent do, we additionally align the concept vectors of the common maze with corresponding ones generated by the first agent (using L2 loss as a training constraint). When the training of all five agents is completed, each agent can successfully navigate through the 3720 mazes it has learned, achieving a near 100% success rate (agent 1: 3710 of 3720; agent 2: 3709 of 3720; agent 3: 3713 of 3720; agent 4: 3713 of 3720; agent 5: 3708 of 3720).

We let each agent navigate in its unseen maze based on the received concept by communication. Due to the choice of an epsilon value of 80% when giving the strategy, we set a threshold with a sufficiently high tolerance, allowing the agent to take a maximum of 100 steps in the maze. If it fails to reach the end point after exceeding 100 steps, it is considered a failure. In fact, almost all of these successful cases were achieved by taking the shortest path. The success rate results are shown in the **Reply Fig. 5**. below, range from 50% to 60%.

We appreciate both two reviewer's suggestion to compare with a stronger baseline, and agree that such a comparison would be valuable. However, since there has been no previous research that achieves 0-shot skill acquisition solely through communication like us (which is also a core concern of MARL in **Reviewer #2, Comment 1**), we had no choice but to use the chance level as the baseline again... In this series of 9x9 mazes,

the average shortest path is between 20 and 30 steps. Let's take the shortest path of 20 steps. Then, the random policy is to choose one action from the four available at each state, which means the probability of choosing the correct action in one step is 0.25. To successfully navigate out of the maze by random policy, the probability is 0.25^{20} , a value infinitely close to 0.

These results highlight that the CATS Net architecture is not limited to learning concepts of visual objects. It can extract and communicate abstract, functional concepts that emerge from task performance in a non-visual domain, even when there are no direct linguistic labels for these emergent concepts.

Reference List:

- Barsalou, L. W. (2003). Abstraction in perceptual symbol systems. *Philosophical Transactions of the Royal Society of London. Series B: Biological Sciences*, 358(1435), 1177–1187. <https://doi.org/10.1098/rstb.2003.1319>
- Bi, Y., Wang, X., & Caramazza, A. (2016). Object Domain and Modality in the Ventral Visual Pathway. *Trends in Cognitive Sciences*, 20(4), 282–290. <https://doi.org/10.1016/j.tics.2016.02.002>
- Binder, J. R., Conant, L. L., Humphries, C. J., Fernandino, L., Simons, S. B., Aguilar, M., & Desai, R. H. (2016). Toward a brain-based componential semantic representation. *Cognitive Neuropsychology*, 33(3–4), 130–174. <https://doi.org/10.1080/02643294.2016.1147426>
- Borghi, A. M., Binkofski, F., Castelfranchi, C., Cimatti, F., Scorolli, C., & Tummolini, L. (2017). The challenge of abstract concepts. *Psychological Bulletin*, 143(3), 263–292. <https://doi.org/10.1037/bul0000089>
- Calzavarini, F. (2024). Rethinking modality-specificity in the cognitive neuroscience of concrete word meaning: A position paper. *Language, Cognition and Neuroscience*, 39(7), 815–837. <https://doi.org/10.1080/23273798.2023.2173789>
- Chiou, R., Humphreys, G. F., Jung, J., & Lambon Ralph, M. A. (2018). Controlled semantic cognition relies upon dynamic and flexible interactions between the executive “semantic control” and hub-and-spoke “semantic representation” systems. *Cortex; a Journal Devoted to the Study of the Nervous System and Behavior*, 103, 100–116. <https://doi.org/10.1016/j.cortex.2018.02.018>
- Contier, O., Baker, C. I., & Hebart, M. N. (2024). Distributed representations of behaviour-derived object dimensions in the human visual system. *Nature Human Behaviour*, 8(11), 2179–2193. <https://doi.org/10.1038/s41562-024-01980-y>
- Diachek, E., Blank, I., Siegelman, M., Affourtit, J., & Fedorenko, E. (2020). The Domain-General Multiple

Demand (MD) Network Does Not Support Core Aspects of Language Comprehension: A Large-Scale fMRI Investigation. *The Journal of Neuroscience: The Official Journal of the Society for Neuroscience*, 40(23), 4536–4550. <https://doi.org/10.1523/JNEUROSCI.2036-19.2020>

Dosovitskiy, A., Beyer, L., Kolesnikov, A., Weissenborn, D., Zhai, X., Unterthiner, T., Dehghani, M., Minderer, M., Heigold, G., Gelly, S., Uszkoreit, J., & Hounsby, N. (2021). *An Image is Worth 16x16 Words: Transformers for Image Recognition at Scale* (No. arXiv:2010.11929). arXiv. <https://doi.org/10.48550/arXiv.2010.11929>

Evans, M., Krieger-Redwood, K., Gonzalez Alam, T. R. J., Smallwood, J., & Jefferies, E. (2020). Controlled semantic summation correlates with intrinsic connectivity between default mode and control networks. *Cortex*, 129, 356–375. <https://doi.org/10.1016/j.cortex.2020.04.032>

Hebart, M. N., Dickter, A. H., Kidder, A., Kwok, W. Y., Corriveau, A., Wicklin, C. V., & Baker, C. I. (2019). THINGS: A database of 1,854 object concepts and more than 26,000 naturalistic object images. *PLOS ONE*, 14(10), e0223792. <https://doi.org/10.1371/journal.pone.0223792>

Hebart, M. N., Zheng, C. Y., Pereira, F., & Baker, C. I. (2020). Revealing the multidimensional mental representations of natural objects underlying human similarity judgements. *Nature Human Behaviour*, 4(11), 1173–1185. <https://doi.org/10.1038/s41562-020-00951-3>

Hoffman, P., McClelland, J. L., & Lambon Ralph, M. A. (2018). Concepts, control, and context: A connectionist account of normal and disordered semantic cognition. *Psychological Review*, 125(3), 293–328. <https://doi.org/10.1037/rev0000094>

Hu, J., Shen, L., & Sun, G. (2018). Squeeze-and-Excitation Networks. *Proceedings of the IEEE Conference on Computer Vision and Pattern Recognition*, 7132–7141. https://openaccess.thecvf.com/content_cvpr_2018/html/Hu_Squeeze-and-

Excitation_Networks_CVPR_2018_paper.html

Jefferies, E., & Lambon Ralph, M. A. (2006). Semantic impairment in stroke aphasia versus semantic dementia:

A case-series comparison. *Brain*, *129*(8), 2132–2147. <https://doi.org/10.1093/brain/awl153>

Kriegeskorte, N., Mur, M., & Bandettini, P. A. (2008). Representational similarity analysis—Connecting the

branches of systems neuroscience. *Frontiers in Systems Neuroscience*, *2*.

<https://doi.org/10.3389/neuro.06.004.2008>

Li, J., Li, D., Savarese, S., & Hoi, S. (2023). BLIP-2: Bootstrapping Language-Image Pre-training with Frozen Image

Encoders and Large Language Models. *Proceedings of the 40th International Conference on Machine*

Learning, 19730–19742. <https://proceedings.mlr.press/v202/li23q.html>

Martinez Pandiani, D. S., Lazzari, N., Erp, M. van, & Presutti, V. (2023). Hypericons for interpretability: Decoding

abstract concepts in visual data. *International Journal of Digital Humanities*, *5*(2), 451–490.

<https://doi.org/10.1007/s42803-023-00077-8>

Nagrani, A., Yang, S., Arnab, A., Jansen, A., Schmid, C., & Sun, C. (2022). *Attention Bottlenecks for Multimodal*

Fusion (No. arXiv:2107.00135). arXiv. <http://arxiv.org/abs/2107.00135>

Nili, H., Wingfield, C., Walther, A., Su, L., Marslen-Wilson, W., & Kriegeskorte, N. (2014). A Toolbox for

Representational Similarity Analysis. *PLOS Computational Biology*, *10*(4), e1003553.

<https://doi.org/10.1371/journal.pcbi.1003553>

Pandiani, D. S. M., & Presutti, V. (2024). *Seeing the Intangible: Survey of Image Classification into High-Level*

and Abstract Categories (No. arXiv:2308.10562). arXiv. <https://doi.org/10.48550/arXiv.2308.10562>

Radford, A., Kim, J. W., Hallacy, C., Ramesh, A., Goh, G., Agarwal, S., Sastry, G., Askell, A., Mishkin, P., Clark, J.,

Krueger, G., & Sutskever, I. (2021). Learning Transferable Visual Models From Natural Language

Supervision. *Proceedings of the 38th International Conference on Machine Learning*, 8748–8763.

<https://proceedings.mlr.press/v139/radford21a.html>

Wang, X., Krieger-Redwood, K., Cui, Y., Smallwood, J., Du, Y., & Jefferies, E. (2024). Macroscale brain states support the control of semantic cognition. *Communications Biology*, 7(1), 926.

<https://doi.org/10.1038/s42003-024-06630-7>

Woo, S., Park, J., Lee, J.-Y., & Kweon, I. S. (2018). CBAM: Convolutional Block Attention Module. *Proceedings of the European Conference on Computer Vision (ECCV)*, 3–19.

https://openaccess.thecvf.com/content_ECCV_2018/html/Sanghyun_Woo_Convolutional_Block_Attention_ECCV_2018_paper.html